# Being More Lightweight and Practical: Mini-sized Contrastive Learning Pre-trained Models for Fine-grained Traffic Task

**Shuhao Li**[1]   **Weidong Yang**[1 2 †]   **Ben Fei**[3]   **Yue Cui**[4]   **Lipeng Ma**[1]   **Fan Zhang**[5]

## Abstract

Fine-grained traffic prediction is critically important for mitigating traffic congestion in key urban areas and for providing lane-change guidance in autonomous vehicles and navigation systems. However, task-specific models are not efficient enough, city-scale pre-trained models often overlook fine-grained requirements, and the demand for extensive computational resources hinders practical deployment. To address this issue, we developed a lightweight pre-training framework, **MiniTraffic**. This framework leverages abundant road-level data to address lane-level data scarcity through a frequency domain stability augmentation module and captures road-lane correlations via contrastive clustering to construct small-scale graph structures, significantly reducing model parameters. Fine-tuning with minimal target data provides a unified and efficient solution for fine-grained traffic prediction. In multi-granularity traffic prediction tasks across six fine-grained datasets, MiniTraffic demonstrated superior performance compared to existing baselines.

## 1. Introduction

Fine-grained traffic prediction encompasses both road-level and lane-level forecasting, offering more detailed data compared to traditional large-scale urban traffic prediction, which is crucial for precise urban management. This data supports not only regional temporary traffic control and traffic signal adjustments (Wu et al., 2025) but also provides accurate information for lane-changing planning and

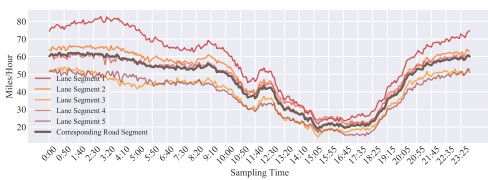

*(a)* The average speed at different times of the day for various lanes and corresponding roads.

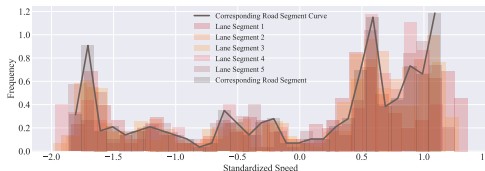

*(b)* The standardized speeds are compared in the frequency domain, with higher overlap areas indicating greater similarity.

*Figure 1.* Comparison between a road segment and its corresponding lanes in the time domain and frequency domain.

guidance in autonomous vehicles and navigation systems (Hu et al., 2023; Qu et al., 2025), significantly enhancing the responsiveness and efficiency of urban traffic systems. Large-scale urban traffic models based on large language models—such as UrbanGPT (Li et al., 2024c), TrafficGPT (Zhang et al., 2024), and ST-LLM (Liu et al., 2024a) or on pre-trained foundational models like TFM (Wang et al., 2023), UniST (Yuan et al., 2024), offer new opportunities for traffic prediction due to their strong generalization capabilities and their ability to handle multiple tasks within a unified framework. However, due to the scarcity of supervised signals in lane-level datasets and the lack of explicit modeling of road–lane associations, these models overlook the specific requirements of fine-grained traffic tasks. This oversight not only limits their practical applicability but also diminishes their effectiveness in specialized scenarios.

As we shift our focus toward more generalized models for fine-grained traffic prediction, it is crucial to identify the key practical challenges that hinder effectiveness and deployment. More specifically, **Effectively handling imbalanced training data remains a critical bottleneck**. Compared to the abundant road-level traffic data, lane-level data is relatively scarce. This imbalance poses significant challenges to developing pre-trained models, as sufficient training data is key to demonstrating strong generalization capa-

---

[1]College of Computer Science and Artificial Intelligence, Fudan University [2]Zhuhai Fudan Innovation Research Institute [3]Department of Information Engineering, The Chinese University of Hong Kong [4]Tongyi Lab, Alibaba Group [5]GZHU-SCHB Intelligent Transportation Joint Lab, Guangzhou University. Correspondence to: Shuhao Li <shli23@m.fudan.edu.cn>, Weidong Yang <wdyang@fudan.edu.cn>.

*Proceedings of the 43rd International Conference on Machine Learning*, Seoul, South Korea. PMLR 306, 2026. Copyright 2026 by the author(s).

bilities (Chen et al., 2023b).To address this, it is crucial to develop strategies that leverage limited and imbalanced data resources to train models capable of adapting to diverse fine-grained traffic prediction tasks. **In addition, fine-grained traffic prediction involves multiple tasks with different requirements and challenges**. Road-level and lane-level predictions are inherently interconnected, with traffic conditions at the road level influencing lane-level behavior and vice versa. Simultaneously modeling both levels not only leverages these correlations for more accurate predictions but also enhances efficiency by reducing redundant computations. Effectively aligning and integrating these tasks is a key challenge in designing a "one-stop" solution, necessitating a flexible and efficient model architecture capable of handling traffic data at multiple granularity levels. **Lastly, model size and computational demand are important issues**. While large traffic models perform well in large-scale urban scenarios, they often require substantial training resources and time (Jin et al., 2021; Duan et al., 2019). Fine-grained traffic prediction for smaller areas demands higher dynamism and variability, necessitating more frequent re-training or fine-tuning. In most practical applications, the computational resources required for training or even fine-tuning large models are often unacceptable, severely limiting the practical application of pre-trained models.

To tackle these challenges, we propose the first mini-sized pre-trained model (MiniTraffic) specifically designed for multi-task processing in fine-grained traffic prediction, leveraging the advantages of pre-trained models to overcome the shortcomings of existing approaches in this field. We conducted a detailed analysis of road-level and lane-level data, revealing differences in traffic speeds between roads and lanes in the PeMS dataset, yet also identifying a certain correlation between them, as shown in **Figure 1(a)**. Although speed values differ, the patterns of change are similar, with **Figure 1(b)** further illustrating the similarity between lane speed and road speed in the frequency domain. Based on this observation, we designed the MiniTraffic framework—a lightweight model with only 100k trainable parameters, capable of being pre-trained on a single A100 GPU. The framework is built to utilize road-level data for pre-training and to transfer this knowledge to lane-level prediction tasks. It includes a frequency domain stability data augmentation module tailored for road data, a pre-training backbone with an attention map contrastive clustering learning module, and fine-tuning strategies for different task granularities. These components fully exploit the correlation between roads and lanes, achieving accurate road-level predictions and effective transfer to lane-level tasks.

In summary, our contributions are as follows: (1) We propose the first pre-training model specifically designed for fine-grained traffic prediction tasks, with a focus on lightweight and practical deployment in real-world scenar-

ios; (2) we design a frequency domain stability augmentation module that simulates multi-lane traffic dynamics from road-level data while preserving trend consistency; (3) we introduce an attention-based contrastive clustering module that captures road-lane correlations, enabling accurate road-level modeling and effective transfer to lane-level tasks; and (4) we conduct extensive experiments across multiple datasets, demonstrating that MiniTraffic consistently outperforms task-specific baselines at corresponding granularities.

## 2. Related Works

**Fine-grained traffic prediction** serves as a foundational task for travel planning and road management, evolving from early lane-level traffic flow theories (Papageorgiou & Schmidt, 1983; Zhang, 1998) to road-level statistical models such as ARIMA (Kumar & Jain, 1999) and Seasonal ARIMA (Kumar & Vanajakshi, 2015), and later to deep learning-based approaches. With the increasing demands of traffic signal control, autonomous driving, and dynamic lane systems, the importance of fine-grained prediction has grown considerably.

In **road-level prediction**, where abundant data is more accessible, rapid progress has been made by treating road segments as basic modeling units (Cui et al., 2023). Spatial modeling has advanced from grid-based CNNs (Zhang et al., 2016; Lv et al., 2018) to graph neural networks (GNNs) (Yu et al., 2018; Wu et al., 2020; Cui et al., 2021; Li et al., 2023b) and attention-based architectures (Guo et al., 2019; Song et al., 2020; Fang et al., 2021). Temporally, RNNs (Li et al., 2018; Jiang et al., 2023), dilated convolutions (Wu et al., 2019), and attention mechanisms (Bai et al., 2020) have all contributed to improving temporal representation. Recently, pre-trained plugin models have also emerged to enhance generalization in road-level prediction(Li et al., 2024d; 2023c).

In contrast, **lane-level prediction** progresses more slowly due to the high cost of sensor deployment and data collection. This task involves modeling lane segments and their interactions within each road segment. Early spatial approaches include grid-based CNNs and intersection-specific lane modeling (Ke et al., 2020; Lu et al., 2020a; Ma et al., 2020), while recent methods have adopted flexible graph structures for more expressive modeling (Li et al., 2024a; Zhou et al., 2022; Li et al., 2023a; Wang et al., 2021). Temporally, lane-level prediction methods often mirror those at the road level. McgVAE (Li et al., 2024b) represents a recent effort to jointly model road- and lane-level data using an ensemble architecture.

However, task-specific or ensemble models often struggle with cross-region transferability. While road-level pre-trained plugin models can enhance generalization across

regions, they face limitations when applied to fine-grained tasks. Therefore, models capable of handling both regional and multi-granularity data are more desirable for supporting real-world traffic management scenarios.

## 3. Problem Formulation

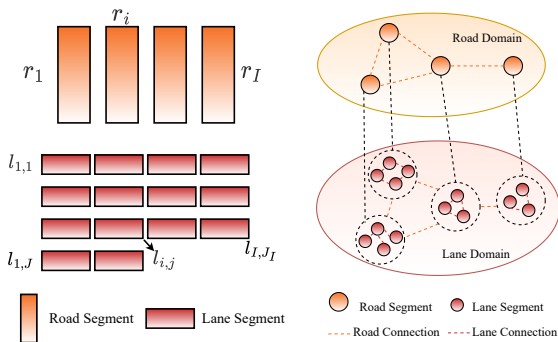

*(a)* Division of road segments and lane segments

*(b)* Cross-domain relationship

*Figure 2.* Correspondence of fine-grained traffic network.

As shown in **Figure 2(a)**, to more accurately describe road segments with irregular numbers of lanes within a road network, we define the road network and lane network as two separate undirected graphs: $G^R = (V^R, E^R, A^R)$ and $G^L = (V^L, E^L, A^L)$. Here, the node $r_i \in V_R$ in $G^R$ represent the $i$-th road segment, while the nodes $l_{i,j}$ in $G^L$ denote the $j$-th lane segment of the $i$-th road segment. The edge $e^{r_i, r_a} \in E^R$ represents the adjacency relationship between road segments $r_i$ and $r_a$, and the edge $e^{l_{i,j}, l_{a,b}} \in E^L$ indicates the adjacency relationship between lane segments $l_{i,j}$ and $l_{a,b}$. $A^R \in \mathbb{R}^{N^R \times N^R}$ and $A^L \in \mathbb{R}^{N^L \times N^L}$ are the adjacency matrices corresponding to the road and lane networks, respectively. Additionally, $N^R = I$ and $N^L = \sum_{i=1}^{I} J_i$, where $J_i$ is the maximum number of lanes for the road segment $r_i$, representing the total number of road segments and lane segments, respectively.

**Problem 1 Domain Transfer Learning.** Given a set of diverse road domain data $G^{Source} = \{G^{R_1}, ..., G^{R_s}\}$, where $s$ represents the number of road domain sources, and lane domain data $G^L$ characterized by data scarcity. In this setting, the model initially undergoes training on $G^{Source}$, leveraging knowledge from multiple road domain datasets, represented as $X^{Source}$. Subsequently, it is tasked with predicting the fine-grained lane domain $G^L$, where $G^L$ possesses only limited structured data available for use. **Figure 2(b)** illustrates the process of knowledge transfer from a single source road domain $G^R$ to the lane domain $G^L$.

Give a time interval $t$, we use $x_t^{r_i}$ to represent the average traffic state of the road segment $r_i$ at time $t$, while $x_t^{l_{i,j}}$ is used to denote the average traffic state of the lane segment $l_{i,j}$ at time $t$. The set $X_t^R = [x_t^{r_1}, x_t^{r_2}, ..., x_t^{r_I}]$ is used to denote the traffic states across all road segments within a road domain $G^R$ at time $t$. Similarly, the set $X_t^L =$

$[x_t^{l_{1,1}}, x_t^{l_{1,2}}, ..., x_t^{l_{I,J_I}}]$ represents the traffic states across all lane segments within the lane domain $G^L$ at the same time.

**Problem 2 Road-Level Traffic Prediction.** Given $X^R$ of window size $T$, $X^R = \{X_1^R, X_2^R, ..., X_T^R\}$, representing the historical states of all nodes in the road domain $G^R$ over the past $T$ time slices, the task is to utilize multi-source road domains $G^{Source}$ and $X^{Source}$ to predict the future traffic states for any target road domain as $\hat{Y}^R = \{X_H^R | H = t+1, \dots, t+h\}$, where $h$ denotes the number of prediction steps. This can be formulated as:

$$f[X^{Source}, G^{Source}, X^R, G^R] \rightarrow \hat{Y}^R. \qquad (1)$$

**Problem 3 Lane-Level Traffic Prediction.** The goal of the lane-level traffic prediction task is to leverage multi-source road domain knowledge $X^{Source}, G^{Source}$, along with few-shot lane domain data for fine-tune, to predict future traffic states in the lane domain as $\hat{Y}^L = \{X_H^L | H = t+1, \dots, t+h\}$. This can be formulated as:

$$f[X^{Source}, G^{Source}, X^L, G^L] \rightarrow \hat{Y}^L. \qquad (2)$$

## 4. MiniTraffic Framework

We present the MiniTraffic framework to capture fine-grained traffic correlations and support efficient transfer across granularities. Based on a unified pre-training paradigm, it enforces spectral consistency, robustness to shifts, and semantic sparsity through frequency-domain regularization, stochastic masking, and contrastive clustering. **Figure 3** shows the pipeline. Pre-training applies spectral perturbations to simulate lane-level variability, with patch partitioning and masking for contextual inference. Contrastive clustering then forms coherent subgraphs for localized attention. The reduction head restores dimensionality. For downstream tasks, granularity-aware fine-tuning adapts road-level inputs via extension and pooling, while lane-level retains spectral augmentation with lightweight heads. This preserves shared knowledge and enables efficient adaptation for real-world deployment, while keeping the framework lightweight and scalable.

### 4.1. Multi-source Pre-training

Effective pre-training is fundamental to MiniTraffic's cross-granularity transfer. We propose a unified paradigm that preserves spectral consistency, enforces robustness, and induces structure-aware sparsity through frequency-domain regularization, stochastic masking, and contrastive clustering. Together, these mechanisms enhance representation learning and enable efficient road-to-lane transfer.

**Frequency Domain Stability Augmentation.** In contrast to prior methods that apply frequency domain augmentation primarily for generic data enhancement ([Kim et al., 2021a](#);

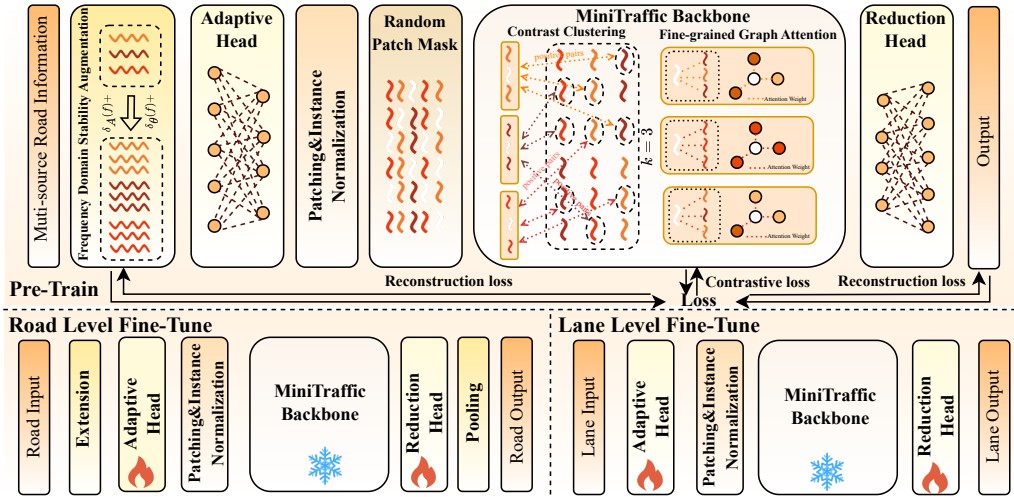

*Figure 3.* The transfer pre-training process of MiniTraffic and its fine-tuning process in road-level and lane-level tasks.

Chen et al., 2023a), we strategically exploit the structural correlations between road-level and lane-level traffic. By introducing task-specific constraints in the frequency domain, we facilitate effective knowledge transfer from road to lane scale, mitigating the scarce lane-level pre-training data limitations. This aligns with the observations in **Figure 1** and is further supported by the physical connection between roads and lanes embodied in the traffic flow conservation law (Shi et al., 2021; Li et al., 2025b); we preserve frequency stability while injecting controlled variations to simulate multi-lane patterns.

We apply a matrix-form Discrete Fourier Transform (DFT) to the road-level state $X^R \in \mathbb{R}^{N^R \times T}$:

$$\tilde{X}^R = X^R \cdot F_T, \quad \text{with } [F_T]_{t,f} = e^{-j\frac{2\pi(t-1)(f-1)}{T}}. \quad (3)$$

Each frequency coefficient is further decomposed into magnitude and phase components:

$$\tilde{X}^R(f) = A(f) \cdot e^{j\theta(f)}, \quad (4)$$

$A(f) = |\tilde{X}^R(f)|$ and $\theta(f) = \arg(\tilde{X}^R(f))$. We perturb the frequency representation to simulate multi-lane variability while preserving spectral consistency. Each complex coefficient is decomposed as $X^R(f) = A(f) \cdot e^{j\theta(f)}$, where $A(f)$ and $\theta(f)$ denote the magnitude and phase. We then inject Gaussian noise: $A(f) + \delta_A(f)$ and $\theta(f) + \delta_\theta(f)$, with $\delta_A \sim \mathcal{N}(0, \sigma_A^2(f))$, $\delta_\theta \sim \mathcal{N}(0, \sigma_\theta^2(f))$. restrict perturbations to informative frequencies, we apply a spectral mask $\Gamma(f) = \mathbb{I}(A(f)^2 > \tau \cdot \max_{f'} A(f')^2)$, where $\tau$ is a learnable parameter optimized during training. This enables the model to focus its augmentation on dominant frequency components that are likely to encode structural patterns. The perturbation magnitude is further controlled by an amplitude threshold $\epsilon(f) = \lambda \cdot \max A(f)$, where $\lambda \in (0,1)$. This ensures the injected noise does not overly distort the signal's

spectral energy. The augmented spectrum is thus:

$$\tilde{X}_d^R(f) = \left( A(f) + \Gamma(f) \cdot \delta_A^{(d)}(f) \right) e^{j(\theta(f) + \Gamma(f) \cdot \delta_\theta^{(d)}(f))}, \quad (5)$$

where $d = 1, \dots, D$, to ensure augmentation stability, we approximate the relative change in spectral energy as $\left| \|\tilde{X}_d^R(f)\|_2^2 - \|X^R(f)\|_2^2 \right| / \|X^R(f)\|_2^2 \lesssim \lambda^2 \cdot \frac{\sum_f \Gamma(f) A(f)^2}{\sum_f A(f)^2}$. This ensures the global frequency structure remains intact. Finally, the perturbed time-domain signals are recovered by inverse DFT:

$$\tilde{X}_d^R = \Re \left( \tilde{X}_d^R(f) \cdot F_T^{-1} \right), \quad (6)$$

where only the real part is retained to ensure compatibility with real-valued inputs. To align the reconstructed signals with the MiniTraffic backbone, we apply an Adaptive Head—a lightweight MLP that projects the augmented output $\tilde{X}^R \in \mathbb{R}^{(N^R \cdot D) \times T}$ into the required shape $\hat{X}^R \in \mathbb{R}^{N' \times T'}$ via $\hat{X}^R = \text{MLP}(\tilde{X}^R)$. This step accommodates source heterogeneity (e.g., road count, segment scale) and ensures temporal alignment before feeding into the unified pre-training encoder.

**Random Patch Mask.** Distributional shift has been shown to significantly affect the generalization performance of prediction models (Kim et al., 2021b). To mitigate such shift, we apply instance normalization to standardize the input statistics at the segment level:

$$\hat{X}_{\text{norm}}^R = \frac{\hat{X}^R - \mu(\hat{X}^R)}{\sigma(\hat{X}^R)}, \quad (7)$$

where $\mu(\cdot)$ and $\sigma(\cdot)$ denote the mean and standard deviation computed per sample and per channel. Following normalization, the temporal dimension of each road segment is divided into non-overlapping patches of fixed length $q$, yielding a

patch tensor $P^R \in \mathbb{R}^{N' \times m \times q}$, where $m = T'/q$. Each patch $p_n^{r_i}$ corresponds to the $n$-th segment of the $i$-th road node.

To enhance robustness and encourage contextual inference, we randomly mask a subset of patches by applying a binary matrix $\mathbf{M} \in \{0,1\}^{N' \times m}$. The masked patch representation is then:

$$p_j^{r_i} \leftarrow \mathbf{M}_{i,j} \cdot p_j^{r_i}, \tag{8}$$

where $\cdot$ denotes element-wise multiplication. This procedure prevents over-reliance on specific local segments and encourages the model to learn redundant and transferable spatio-temporal patterns.

**MiniTraffic Backbone.** To capture fine-grained temporal similarity among lane-level segments, we design a contrastive clustering approach that not only reduces computational scale but also constructs semantic-driven patch graphs to preserve long-range dependencies and spectral fidelity. The overall process consists of two stages: (1) learning patch similarity via contrastive learning, and (2) constructing dynamic sparse graphs for graph attention propagation.

*Contrastive Patch Similarity Learning.* Given the normalized and patchified input $P^R \in \mathbb{R}^{N' \times m \times q}$, we flatten it into a set of patch embeddings $\{p_i\}_{i=1}^{N' \cdot m}$, where each $p_i \in \mathbb{R}^q$. For each pair of visible (unmasked) patches $p_i, p_j$, we compute the cosine similarity:

$$s_{ij} = \frac{p_i^\top p_j}{\|p_i\| \|p_j\|}. \tag{9}$$

To optimize patch representations, we formulate the contrastive objective based on the InfoNCE loss. Let $p_i^+$ be a positive pair (e.g., same road, same time slice), and $\mathcal{N}_i$ be the set of negatives. The similarity matrix is computed as:

$$\mathcal{S}_{ij} = \exp\left(\frac{s_{ij}}{\ell}\right), \quad \text{with temperature } \Upsilon > 0. \tag{10}$$

The contrastive objective is:

$$\mathcal{L}_{\text{CL}} = -\sum_{i=1}^{N'} \log \frac{\mathcal{S}_{i,i^+}}{\sum_{j \in \mathcal{N}_i \cup \{i^+\}} \mathcal{S}_{i,j}}. \tag{11}$$

This loss encourages patches with similar temporal dynamics to have aligned representations, while pushing apart unrelated ones.

*Clustering-Based Graph Construction.* Once similarity is optimized, we construct a localized graph structure where each node corresponds to a patch embedding. For each patch $p_i$, we identify its $k$-nearest neighbors to form a sparse adjacency matrix $\widetilde{A} \in \mathbb{R}^{(N' \cdot m) \times (N' \cdot m)}$:

$$\widetilde{A}_{i,j} = \begin{cases} 1, & \text{if } j \in \mathcal{N}_k(i) \\ 1, & \text{if } i \text{ is masked and } j \in \mathcal{N}_{2k}(adj(I)). \\ 0, & \text{otherwise} \end{cases} \tag{12}$$

Here, $\mathcal{N}_k(i)$ denotes the $k$-nearest neighbors based on learned similarity (Liu et al., 2024b), and $adj(i)$ refers to adjacent (non-masked) patches. This graph is symmetric and localized, significantly reducing computational cost compared to global spatio-temporal graphs. Since $\widetilde{A}$ is built from semantic similarity rather than spatial proximity, patches with aligned dynamics can be directly connected across long distances, preserving long-range dependencies without dense global attention.

*Fine-grained Graph Attention.* To further refine patch representations within localized contexts, we apply a fine-grained graph attention mechanism on the sparse graphs constructed via contrastive clustering. Unlike traditional attention methods that operate over fully connected graphs or global spatio-temporal matrices, our attention is constrained to patch-wise neighborhoods defined by $\widetilde{A} \in \mathbb{R}^{(N' \cdot m) \times (N' \cdot m)}$, which significantly reduces computational overhead. Let $p_i$ denote the embedding of the $i$-th patch, and $\mathcal{N}_i$ its neighbor set from $\widetilde{A}$. For each node pair $(i, j)$ within $\mathcal{N}_i$, we compute the attention coefficient as:

$$e_{ij} = \text{LeakyReLU}(\mathbf{a}^\top [\mathbf{W} p_i \| \mathbf{W} p_j]), \tag{13}$$

where $\mathbf{W}$ is a shared linear transformation, and $\|$ denotes vector concatenation. The attention weights are then normalized locally:

$$\alpha_{ij} = \frac{\exp(e_{ij})}{\sum_{k \in \mathcal{N}_i} \exp(e_{ik})}. \tag{14}$$

Each node aggregates its neighborhood information via a weighted sum:

$$\bar{p}_i = \sigma\left(\sum_{j \in \mathcal{N}_i} \alpha_{ij} \cdot \mathbf{W} p_j\right), \tag{15}$$

where $\bar{p}_i$ is the updated patch representation, and $\sigma(\cdot)$ is a ReLU nonlinearity. Compared to conventional full-graph GAT, our localized formulation reduces the number of attention computations from $\mathcal{O}((N' \cdot m)^2)$ to $\mathcal{O}(k \cdot N' \cdot m)$, where $k \ll N' \cdot m$ is the sparsity parameter, with $k$ proportional to the mask ratio ($k = 10r$) to preserve context. This sparsity is task-aware, as $\widetilde{A}$ is dynamically constructed from patch-level semantic similarity rather than fixed spatial proximity.

Finally, the attended patch embeddings $\bar{P}^R$ are aggregated and decoded using a lightweight MLP to recover the temporal structure. We apply instance-wise de-normalization to produce the final output:

$$\ddot{X} = \text{MLP}(\bar{P}^R) \cdot \sigma(\hat{X}^R) + \mu(\hat{X}^R). \tag{16}$$

**Pre-training Objective.** During the pre-training phase, training is conducted through a combination of reconstruc-

tion loss and contrastive loss:

$$\mathcal{L} = \frac{1}{N^R \times D} \sum_{i=1}^{N^R \times D} (\ddot{x}^{r_i} - \hat{x}^{r_i})^2 + \mathcal{L}_{cl}. \quad (17)$$

### 4.2. Granularity-Aware Fine-Tuning

To adapt MiniTraffic to downstream prediction tasks at different spatial granularities, we propose a unified fine-tuning strategy that preserves pre-trained knowledge while enabling targeted adaptation. The MiniTraffic Backbone is frozen during fine-tuning, while lightweight task-specific modules are retrained for each granularity.

**Road-Level Fine-Tuning.** For road-level tasks, we replace the pre-trained Frequency Domain Stability Augmentation (FDA) module with a simple Extension module and append a Pooling module to restore output dimensionality. This design maintains compatibility with the pre-training interface while avoiding unnecessary perturbations. Given the road-level input $X^R \in \mathbb{R}^{N^R \times T}$, the Extension module duplicates each road segment representation $D$ times along the spatial axis to match the pre-trained input structure:

$$\hat{X}^R = \text{Repeat}(X^R, D), \quad \hat{X}^R \in \mathbb{R}^{(D \cdot N^R) \times T}. \quad (18)$$

After obtaining the fine-grained outputs $Y \in \mathbb{R}^{(D \cdot N^R) \times H}$ from the Reduction Head, we perform a structure-aware pooling operation to restore the original road-level spatial resolution. Specifically, for each road segment $r_i$, its final representation $\hat{y}_i$ is computed as the average of its $D$ repeated embeddings along the entity axis:

$$\hat{y}_i = \frac{1}{D} \sum_{d=1}^{D} Y_{(i-1) \cdot D + d}, \quad i = 1, \ldots, N^R. \quad (19)$$

This yields the compact prediction matrix $\hat{Y} = [\hat{y}_1, \hat{y}_2, \ldots, \hat{y}_{N^R}]^\top \in \mathbb{R}^{N^R \times H}$. This spatial-entity-wise pooling serves two roles: (1) it semantically integrates redundant views of the same road entity generated for pre-training compatibility, and (2) it enforces invariance across repeated structural tokens, akin to a cross-view consistency constraint.

**Lane-Level Fine-Tuning.** For lane-level tasks, the pre-trained FDA module is retained, as its perturbation design has already modeled lane-level signal variability. In this case, the backbone is also frozen, and only the Adaptive Head and Reduction Head are retrained on limited lane-specific data. This fine-tuning strategy supports few-shot generalization to new lane-level domains.

## 5. Experiment

We pre-trained MiniTraffic using publicly available datasets for road-level prediction and fine-tuned it with benchmark datasets for fine-grained prediction. We compared Mini-Traffic with baseline models across two granular tasks and conducted experiments to validate its performance under various conditions. Additional analyses, including **fine-tuning data ratio**, **cost**, **trainable parameters vs. performance**, **lane-to-road relationship**, and **parameter $\lambda$ study**, are provided in **Appendix A - E**.

### 5.1. Datasets and Experiment Setup

**Datasets Description and Metrics.** We used eight traffic datasets of varying granularity, with five road-level datasets for pre-training and six mixed-granularity datasets for fine-tuning and testing. Their descriptions and statistics are detailed in the **Appendix H,I**, specifically: **1) Pre-training Datasets:** METR-LA, PeMS-Bay (Li et al., 2018; Chen et al., 2024), PeMS-(Road) [1], PeMSF-(Road), and HuaNan-(Road) (Li et al., 2025a; 2024a). **2) Fine-tuning and Testing Datasets:** We conducted comparative experiments on six fine-grained traffic task datasets: PeMS-(Lane/Road), PeMSF-(Lane/Road), and HuaNan-(Lane/Road). We used three evaluation metrics: Mean Absolute Error (MAE), Root Mean Square Error (RMSE), and Mean Absolute Percentage Error (MAPE) to assess performance.

**Baselines Description.** We compared our model with 29 baselines, including 14 lane-level models, 12 road-level models, the only existing fine-grained multi-task model McgVAE (Li et al., 2024b), and two pre-trained plugin model FlashST (Li et al., 2024d), GPT-ST (Li et al., 2023c). **1)Lane-level:** Cat-RF-LSTM (Zhao & Chen, 2022), CEEMDAN-XGBoos (Lu et al., 2020b), LSTM (Graves, 2012), GRU (Cho et al., 2014), FDL (Gu et al., 2019), TM-CNN (Ke et al., 2020), MDL (Lu et al., 2020a), CNN-LSTM (Ma et al., 2020), HGCN (Zhou et al., 2022), GCN-GRU (Li et al., 2023a), ST-AFN (Shen et al., 2021), STA-ED (Zheng et al., 2022), STMGG (Wang et al., 2021), ST-ABC (Li et al., 2024a). **2)Road-level:** DCRNN (Li et al., 2018), STGCN (Yu et al., 2018), MTGNN (Wu et al., 2020), ASTGCN (Guo et al., 2019), GraphWaveNe (Wu et al., 2019), STSGCN (Song et al., 2020), AGCRN (Bai et al., 2020), STGODE (Fang et al., 2021), STAEformer (Liu et al., 2023a), MegaCRN (Jiang et al., 2023), Time-Mixer (Wang et al., 2024), iTransformer(Liu et al., 2023c).

Grid-based models like TM-CNN, MDL, and CNN-LSTM cannot handle the PeMSF-Lane datasets due to irregular lane counts, so they were only compared on the PeMS-Lane and HuaNan-Lane datasets. Descriptions and reproduction details are provided in the **Appendix J**.

**Experimental Setups.** Experiments were conducted on a platform with an Intel Xeon CPU Max 9462 processor (2.70 GHz) and six NVIDIA A100 80GB SXM GPUs. The input

---

[1]https://pems.dot.ca.gov/

*Table 1.* Comparison on PeMS-(Lane/Road) Datasets and HuaNan-(Lane/Road) Datasets, [†] denotes multi-task models, and [*] indicates pre-trained models.

| Tasks | Dataset | PeMS | | | | | | | | | HuaNan | | | | | | | | |
|---|---|---|---|---|---|---|---|---|---|---|---|---|---|---|---|---|---|---|---|
| | Horizon | 3 | | | 6 | | | 12 | | | 3 | | | 6 | | | 12 | | |
| | Metric | MAE | RMSE | MAPE | MAE | RMSE | MAPE | MAE | RMSE | MAPE | MAE | RMSE | MAPE | MAE | RMSE | MAPE | MAE | RMSE | MAPE |
| Lane-Task | Cat-RF-LSTM | 7.83 | 10.79 | 82.32% | 7.89 | 11.11 | 82.74% | 8.38 | 12.02 | 84.31% | 18.35 | 22.93 | 94.43% | 18.48 | 23.11 | 94.61% | 18.86 | 23.44 | 100.22% |
| | CEEMDAN-XGBoost | 7.33 | 10.07 | 75.52% | 7.36 | 10.32 | 75.78% | 8.01 | 10.62 | 78.12% | 16.84 | 21.03 | 80.50% | 16.91 | 21.36 | 79.68% | 17.51 | 22.29 | 84.65% |
| | LSTM | 7.04 | 9.84 | 54.38% | 7.37 | 10.36 | 56.20% | 7.85 | 10.97 | 58.45% | 16.87 | 21.51 | 80.18% | 16.95 | 21.69 | 81.32% | 17.04 | 21.91 | 81.78% |
| | GRU | 6.68 | 9.42 | 43.22% | 7.00 | 10.00 | 44.63% | 7.64 | 10.73 | 46.67% | 16.25 | 20.82 | 72.07% | 16.39 | 21.11 | 73.38% | 16.43 | 21.13 | 74.04% |
| | FDL | 6.86 | 9.68 | 44.00% | 7.27 | 10.28 | 45.66% | 7.92 | 11.13 | 47.81% | 16.52 | 21.19 | 73.20% | 16.90 | 21.79 | 75.84% | 16.90 | 21.91 | 76.97% |
| | TM-CNN | 4.87 | 7.69 | 23.66% | 5.17 | 8.32 | 25.04% | 5.97 | 9.58 | 29.30% | 13.95 | 19.11 | 52.25% | 14.40 | 19.95 | 57.35% | 16.06 | 21.75 | 76.19% |
| | MDL | 4.35 | 7.09 | 21.78% | 4.97 | 8.11 | 24.69% | 5.66 | 9.21 | 29.03% | 4.92 | 7.96 | 19.45% | | 9.19 | 22.22% | 8.54 | 12.81 | 30.57% |
| | CNN-LSTM | 8.39 | 11.60 | 85.15% | 8.52 | 11.74 | 87.63% | 8.63 | 12.04 | 89.86% | 12.27 | 17.97 | 62.74% | 12.53 | 18.09 | 65.27% | 13.52 | 19.09 | 72.50% |
| | HGCN | 4.77 | 7.72 | 24.27% | 5.27 | 8.57 | 26.29% | 5.98 | 9.72 | 30.24% | 11.11 | 15.62 | 35.08% | 11.35 | 15.84 | 36.40% | 11.78 | 16.28 | 38.09% |
| | GCN-GRU | 4.75 | 7.72 | 23.59% | 5.21 | 8.54 | 26.12% | 6.20 | 9.91 | 32.12% | 11.09 | 15.54 | 34.93% | 11.39 | 15.91 | 36.66% | 11.85 | 16.40 | 38.08% |
| | ST-AFN | 4.52 | 7.50 | 21.81% | 5.22 | 8.53 | 26.04% | 6.14 | 10.07 | 29.67% | 5.21 | 9.38 | 20.31% | 7.74 | 12.46 | 27.95% | 9.26 | 14.10 | 31.15% |
| | STA-ED | 6.91 | 9.88 | 44.85% | 7.52 | 10.67 | 47.49% | 7.85 | 11.09 | 47.83% | 13.69 | 18.54 | 62.90% | 14.44 | 19.16 | 65.11% | 15.13 | 19.76 | 67.89% |
| | STMGG | 7.00 | 9.19 | 43.18% | 7.82 | 10.69 | 52.52% | 7.95 | 11.25 | 56.02% | 14.38 | 18.06 | 70.35% | 15.21 | 19.22 | 74.51% | 16.21 | 20.04 | 76.37% |
| | ST-ABC† | 4.39 | 6.91 | 20.85% | 4.94 | 7.85 | 25.28% | 5.78 | 8.96 | 28.96% | 4.90 | 7.63 | 19.59% | 5.98 | 9.36 | 23.48% | 8.36 | 12.60 | 30.97% |
| | McgVAE† | 4.34 | 6.72 | 20.66% | 4.89 | 7.71 | 24.36% | 4.58 | 8.92 | 26.20% | 4.58 | 6.28 | 19.24% | 5.72 | | 22.88% | 6.94 | 10.95 | 30.07% |
| | **MiniTraffic*,†** | **3.51** | **5.96** | **18.32%** | **3.94** | **6.83** | **20.24%** | **5.12** | **7.42** | **24.35%** | **3.69** | **5.56** | **18.68%** | **4.11** | **7.76** | **20.81%** | **5.27** | **8.22** | **26.53%** |
| Road-Task | DCRNN | 4.30 | 7.55 | 23.00% | 6.18 | 9.93 | 30.66% | 7.17 | 11.26 | 35.08% | 6.49 | 9.12 | 23.89% | 9.05 | 13.36 | 35.29% | 9.78 | 12.25 | 35.01% |
| | STGCN | 4.21 | 8.14 | 22.08% | 5.60 | 10.17 | 28.58% | 6.79 | 12.36 | 36.48% | 6.42 | 9.11 | 23.33% | 8.46 | 12.83 | 30.19% | 9.12 | 11.94 | 26.67% |
| | MTGNN | 3.14 | 5.56 | 13.88% | 3.72 | 6.69 | 17.48% | 4.69 | 8.20 | 21.82% | 4.66 | 7.32 | 14.52% | 5.30 | 7.85 | 16.37% | 7.29 | 10.43 | 22.39% |
| | ASTGCN | 4.06 | 6.87 | 17.95% | 5.22 | 8.74 | 24.92% | 6.46 | 10.35 | 32.80% | 5.75 | 8.05 | 19.57% | 7.70 | 11.16 | 26.61% | 8.99 | 11.25 | 28.18% |
| | GraphWaveNet | 3.27 | 5.76 | 14.23% | 3.97 | 7.01 | 16.65% | 5.20 | 8.98 | 23.86% | 4.81 | 7.56 | 15.04% | 6.88 | 11.20 | 21.74% | 7.41 | 10.41 | 21.91% |
| | STSGCN | 3.23 | 5.59 | 14.06% | 4.03 | 6.87 | 17.63% | 4.88 | 8.45 | 21.86% | 5.09 | 7.60 | 15.57% | 7.12 | 10.89 | 21.27% | 7.54 | 9.83 | 21.69% |
| | AGCRN | 3.22 | 5.57 | 14.16% | 3.98 | 6.81 | 17.39% | 4.87 | 8.40 | 22.13% | 3.86 | 6.05 | 14.01% | 6.38 | 10.02 | 22.80% | 7.83 | 12.40 | 24.08% |
| | STGODE | 3.67 | 6.97 | 16.81% | 5.14 | 9.22 | 26.87% | 7.39 | 12.75 | 38.13% | 4.91 | 7.39 | 14.94% | 6.74 | 10.40 | 19.89% | 7.15 | 9.38 | 20.07% |
| | STAEformer | 3.57 | 6.50 | 15.10% | 4.54 | 7.40 | 19.18% | 5.41 | 9.81 | 23.60% | 4.40 | 6.57 | 15.46% | 7.32 | 10.44 | 26.49% | 8.95 | 13.45 | 26.90% |
| | MegaCRN | 3.25 | 6.08 | 16.07% | 4.31 | 7.74 | 21.52% | 5.33 | 9.52 | 26.36% | 6.16 | 9.12 | 17.84% | 8.63 | 13.57 | 25.47% | 9.67 | 13.29 | 28.63% |
| | Time-Mixer | 3.25 | 5.62 | 14.30% | 4.02 | 6.88 | 17.56% | 4.92 | 8.48 | 22.36% | 3.90 | 6.11 | 14.15% | 6.45 | 10.12 | 23.03% | 7.91 | 12.53 | 24.32% |
| | iTransformer | 3.15 | 5.46 | 13.89% | 3.90 | 6.68 | 17.05% | 4.78 | 8.23 | 21.71% | 3.88 | 6.08 | 14.09% | 6.42 | 10.08 | 22.93% | 7.87 | 12.48 | 24.22% |
| | FlashST* | 3.12 | 5.31 | 12.43% | 3.76 | 6.73 | 15.75% | 4.65 | 8.11 | 20.57% | 4.47 | 6.98 | 14.36% | 5.61 | 7.89 | 15.74% | 7.23 | 10.31 | 22.14% |
| | McgVAE† | 3.23 | 5.38 | 12.59% | 3.70 | 6.68 | 15.60% | 4.64 | 8.16 | 20.20% | 3.68 | 5.19 | 12.09% | 5.28 | 7.46 | 15.06% | 5.52 | 8.19 | 19.95% |
| | GPT-ST* | 3.14 | 5.35 | 12.43% | 3.72 | 6.71 | 15.72% | 4.62 | 7.81 | 19.52% | 4.58 | 7.26 | 14.38% | 6.60 | 10.75 | 20.79% | 7.10 | 9.99 | 20.95% |
| | **MiniTraffic*,†** | **2.33** | **4.23** | **10.01%** | **3.12** | **4.90** | **11.22%** | **3.77** | **5.90** | **14.67%** | **2.71** | **4.26** | **9.89%** | **3.16** | **4.46** | **12.45%** | **4.85** | **7.74** | **17.94%** |

window included 18 timestamps, with prediction horizons set to 3, 6, and 12 timestamps. The mask rate was 0.4, and the patch length was 3. Training was done using the Adam optimizer, with up to 1000 iterations, and early stopping was employed to prevent overfitting. The related code, datasets, and pre-trained models are available[2].

## 5.2. Main Experiment

We compared the fine-tuned performance of MiniTraffic with lane-level and road-level baseline models across six datasets (PeMS-Lane, PeMS-Road, PeMSF-Lane, PeMSF-Road, HuaNan-Lane, HuaNan-Road). The best results are in **bold**, and the second-best results are underlined.

**Table 1** shows MiniTraffic's performance on PeMS-Lane, PeMS-Road, HuaNan-Lane, and HuaNan-Road. MiniTraffic significantly outperforms baseline models in lane-level prediction, reducing MAE by 7%-24% on PeMS-Lane and by 24%-39% on HuaNan-Lane compared to the best Mcg-VAE model. In road-level prediction, while baseline models like FlashST, McgVAE, and GPT-ST perform well, Mini-Traffic shows significant advantages, attributed to its pre-training phase's use of contrastive clustering to capture critical information.

**Table 3** shows the performance on PeMSF-Lane and PeMSF-Road, which have irregular lane counts. MiniTraffic surpasses the best baseline McgVAE on PeMSF-Lane, despite slightly higher errors compared to PeMS-Lane due to increased complexity. On PeMSF-Road, MiniTraffic

achieves the best performance, followed by GPT-ST, demonstrating that the granularity of pre-training data influences performance preferences when predicting across domains.

*Table 2.* Influence of Pre-training Strategies

| | Horizon | PeMS-Lane | | | HuaNan-Lane | | | PeMSF-Lane | | |
|---|---|---|---|---|---|---|---|---|---|---|
| | | MAE | RMSE | MAPE | MAE | RMSE | MAPE | MAE | RMSE | MAPE |
| Domain | 3 | 3.31 | 5.47 | 18.03% | 3.48 | 5.10 | 18.39% | 3.11 | 5.27 | 18.21% |
| | 6 | 3.71 | 6.27 | 19.92% | 3.87 | 7.12 | 20.47% | 3.66 | 6.22 | 20.28% |
| | 12 | 4.82 | 6.81 | 23.96% | 4.96 | 7.54 | 26.11% | 4.75 | 6.94 | 25.18% |
| | | PeMS-Lane | | | HuaNan-Road | | | PeMSF-Road | | |
| | | MAE | RMSE | MAPE | MAE | RMSE | MAPE | MAE | RMSE | MAPE |
| Individual | 3 | 4.33 | 7.15 | 23.54% | 3.55 | 7.33 | 12.65% | 3.08 | 5.60 | 12.79% |
| | 6 | 4.85 | 8.20 | 26.01% | 4.14 | 7.69 | 15.93% | 3.97 | 6.77 | 15.49% |
| | 12 | 6.31 | 8.91 | 31.29% | 6.35 | 13.34 | 22.96% | 4.87 | 7.69 | 19.74% |

## 5.3. Pre-training Strategies Study

To investigate how different pre-training data sources affect the performance of MiniTraffic, we compared under two pre-training mechanisms: one using lane-level data sources for pre-training and the other using a single dataset. **Table 2** shows that pre-training with lane-level data improves performance on PeMS-Lane, HuaNan-Lane, and PeMSF-Lane. This suggests that pre-training on extensive lane-level data can facilitate the learning of lane-level patterns, thereby enhancing accuracy. Conversely, pre-training with a single dataset results in poorer performance, especially on road-level datasets like HuaNan-Road and PeMSF-Road, highlighting the importance of multi-source pre-training.

## 5.4. Ablation Study

We conducted ablation experiments by removing the Frequency Domain Stability Augmentation during pre-training, the Extension and Pooling modules during road-level fine-

---

[2] https://github.com/ShuhaoLii/Mini-Traffic

*Table 3.* Comparison on PeMSF-(Lane/Road) Datasets, [†] denotes multi-task models, and [*] indicates pre-trained models.

| Task | Lane-Task | | | | | | | | | | Road-Task | | | | | | | | | |
|---|---|---|---|---|---|---|---|---|---|---|---|---|---|---|---|---|---|---|---|
| Horizon | 3 | | | 6 | | | 12 | | | | 3 | | | 6 | | | 12 | | |
| Metric | MAE | RMSE | MAPE | MAE | RMSE | MAPE | MAE | RMSE | MAPE | | MAE | RMSE | MAPE | MAE | RMSE | MAPE | MAE | RMSE | MAPE |
| Cat-RF-LSTM | 7.96 | 10.99 | 83.36% | 7.92 | 11.13 | 83.35% | 8.52 | 12.24 | 85.38% | DCRNN | 4.53 | 6.89 | 18.90% | 5.83 | 8.85 | 24.13% | 6.86 | 10.41 | 29.72% |
| CEEMDAN-XGBoost | 7.29 | 9.99 | 74.52% | 7.55 | 10.62 | 78.34% | 7.97 | 10.53 | 77.09% | STGCN | 4.21 | 7.11 | 19.32% | 5.42 | 9.13 | 24.67% | 6.38 | 10.74 | 30.38% |
| LSTM | 7.06 | 9.86 | 55.72% | 7.49 | 10.55 | 55.95% | 7.88 | 10.99 | 59.88% | MTGNN | 3.08 | 5.36 | 13.24% | 3.67 | 6.49 | 16.14% | 4.55 | 8.01 | 21.40% |
| GRU | 6.71 | 9.49 | 44.57% | 7.18 | 10.13 | 44.15% | 7.68 | 10.81 | 48.13% | ASTGCN | 3.92 | 6.15 | 16.74% | 5.05 | 7.90 | 21.37% | 5.94 | 9.29 | 26.33% |
| FDL | 7.03 | 10.03 | 44.97% | 7.24 | 10.12 | 45.57% | 8.11 | 11.53 | 48.87% | GraphWaveNet | 2.98 | 5.08 | 11.18% | 3.70 | 6.04 | 13.01% | 4.36 | 7.30 | 16.03% |
| HGCN | 4.89 | 7.88 | 24.30% | 5.25 | 8.57 | 26.56% | 6.12 | 9.91 | 30.54% | STSGCN | 3.55 | 5.73 | 15.34% | 4.57 | 7.36 | 19.59% | 5.38 | 8.65 | 24.13% |
| GCN-GRU | 4.89 | 8.06 | 24.27% | 5.15 | 8.38 | 25.91% | 6.39 | 10.34 | 33.04% | AGCRN | 3.11 | 5.30 | 13.17% | 4.00 | 6.80 | 16.82% | 4.71 | 8.00 | 20.72% |
| ST-AFN | 4.73 | 7.79 | 24.25% | 5.08 | 8.37 | 23.80% | 6.44 | 10.46 | 33.13% | STGODE | 3.21 | 5.14 | 12.01% | 4.14 | 6.60 | 15.34% | 4.87 | 7.77 | 18.89% |
| STA-ED | 7.21 | 10.35 | 46.98% | 7.35 | 10.39 | 46.25% | 8.19 | 11.61 | 50.10% | STAEformer | 3.38 | 6.07 | 13.77% | 4.47 | 7.25 | 18.19% | 5.13 | 9.16 | 21.65% |
| STMGG | 7.06 | 9.17 | 43.68% | 7.91 | 10.93 | 52.96% | 8.02 | 11.22 | 56.67% | MegaCRN | 3.82 | 6.44 | 16.47% | 4.92 | 8.26 | 21.03% | 5.79 | 9.72 | 25.90% |
| ST-ABC | 4.80 | 6.95 | 25.54% | 5.02 | 7.72 | 33.21% | 6.08 | 9.54 | 30.14% | Time-Mixer | 3.08 | 5.25 | 13.04% | 3.96 | 6.74 | 16.65% | 4.66 | 7.92 | 20.51% |
| | | | | | | | | | | iTransformer | 3.02 | 5.15 | 12.80% | 3.89 | 6.61 | 16.35% | 4.58 | 7.78 | 20.14% |
| | | | | | | | | | | FlashST* | 2.96 | 5.13 | 13.10% | 3.61 | 6.11 | 14.64% | 4.52 | 7.92 | 19.16% |
| McgVAE[†] | 4.42 | 6.92 | 23.35% | 4.40 | 7.62 | 22.52% | 5.97 | 8.40 | 29.51% | McgVAE[†] | 2.96 | 5.04 | 10.22% | 3.58 | 6.03 | 12.32% | 4.24 | 7.26 | 15.94% |
| | | | | | | | | | | GPT-ST* | 2.87 | 4.98 | 10.16% | 3.56 | 5.92 | 12.28% | 4.20 | 7.16 | 15.53% |
| **MiniTraffic***,[†] | 3.52 | 5.99 | 19.30% | 4.15 | 7.07 | 21.49% | 5.38 | 7.89 | 26.68% | **MiniTraffic***,[†] | 2.46 | 4.32 | 9.71% | 3.17 | 5.21 | 11.76% | 3.89 | 5.92 | 14.99% |

*Table 4.* Ablation Study

| Variants | Datasets | PeMSF-Lane | | HuaNan-Lane | | HuaNan-Road | |
|---|---|---|---|---|---|---|---|
| | Metrics | w/o | original | w/o | original | w/o | original |
| Augmen- -tation | MAE | 4.64 | 4.15 | 4.48 | 4.11 | 3.26 | 3.16 |
| | RMSE | 7.88 | 7.07 | 8.46 | 7.76 | 4.61 | 4.46 |
| | MAPE | 23.75% | 21.49% | 22.68% | 20.81% | 12.85% | 12.45% |
| Variants | Datasets | PeMS-Road | | PeMSF-Road | | HuaNan-Road | |
| | Metrics | w/o | original | w/o | original | w/o | original |
| Extension &Pooling | MAE | 3.40 | 3.12 | 3.44 | 3.17 | 3.44 | 3.16 |
| | RMSE | 5.36 | 4.90 | 5.65 | 5.21 | 5.75 | 4.46 |
| | MAPE | 12.29% | 11.22% | 12.76% | 11.76% | 13.53% | 12.45% |
| Variants | Datasets | PeMS-Lane | | PeMSF-Lane | | HuaNan-Road | |
| | Metrics | r/p | original | r/p | original | r/p | original |
| Contrastive Clustering | MAE | 4.93 | 3.94 | 5.11 | 4.15 | 3.23 | 3.16 |
| | RMSE | 8.54 | 6.83 | 8.67 | 7.07 | 4.56 | 4.46 |
| | MAPE | 25.30% | 20.24% | 26.13% | 21.49% | 12.75% | 12.45% |

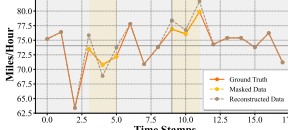

*(a)* Pretrain on PeMS_Road dataset

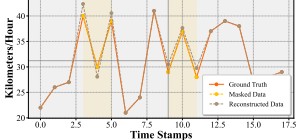

*(b)* Pretrain on HuaNan_Road dataset

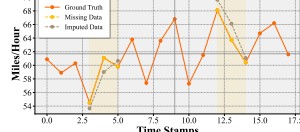

*(c)* Imputation on PeMS_Lane dataset

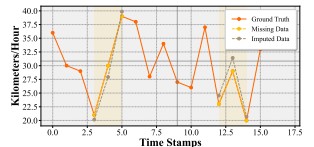

*(d)* Imputation on HuaNan_Lane dataset

*Figure 4.* Reconstruction and imputation performance.

tuning, and replacing Contrastive Clustering (CC) with Full Graph Attention. **Table 4** shows that removing FDA significantly decreases accuracy on PeMSF-Lane and HuaNan-Lane (MAE +9%–11%), with errors also increasing on HuaNan-Road, indicating road-level prediction benefits from this augmentation. Eliminating Extension and Pooling likewise raises errors, especially on HuaNan-Road, likely due to insufficient expansion reducing similarity among positive pairs. Further, removing CC degrades lane-level performance (e.g., PeMSF-Lane MAE 4.15→5.11, MAPE +4.6%) and increases parameters from 119K to 820K, confirming CC both preserves expressiveness and compresses computation.

## 5.5. Parameter Experiment

*Table 5.* Influence of Mask Ratio

| Mask | PeMSF-Lane | | | HuaNan-Lane | | | HuaNan-Road | | |
|---|---|---|---|---|---|---|---|---|---|
| Ratio | MAE | RMSE | MAPE | MAE | RMSE | MAPE | MAE | RMSE | MAPE |
| 10% | 4.60 | 7.93 | 24.19% | 4.56 | 8.71 | 23.43% | 3.51 | 5.01 | 14.01% |
| 20% | 4.23 | 7.19 | 22.32% | 4.21 | 7.91 | 21.62% | 3.25 | 4.55 | 12.93% |
| 30% | 4.19 | 7.11 | 22.10% | 4.18 | 7.84 | 21.41% | 3.22 | 4.52 | 12.80% |
| 40% | 4.15 | 7.07 | 21.49% | 4.11 | 7.76 | 20.81% | 3.16 | 4.46 | 12.45% |
| 50% | 4.21 | 7.14 | 22.25% | 4.17 | 7.83 | 21.55% | 3.20 | 4.50 | 12.89% |
| 60% | 4.32 | 7.34 | 22.95% | 4.26 | 8.04 | 22.23% | 3.27 | 4.61 | 13.29% |

We compared the impact of the patch mask rate during pre-training on subsequent fine-tuning and prediction across PeMSF-Lane, HuaNan-Lane, PeMSF-Road, and HuaNan-

Road datasets. **Table 5** shows that the mask rate significantly affects prediction performance. The best pre-training results were achieved with a 40% mask rate, resulting in the lowest prediction errors. Deviations from 40% increased prediction errors, as an appropriate mask rate enhances generalization by increasing reconstruction difficulty. Too low or too high mask rates reduce this effect. These experimental results demonstrate that selecting an appropriate mask rate is crucial for improving the performance of the MiniTraffic model in both pre-training and downstream tasks.

## 5.6. Pre-training and Imputation Task

**Figures 4(a)** and **4(b)** demonstrate MiniTraffic's reconstruction capability on the PeMS-Road and HuaNan-Road datasets after pre-training by reconstructing real values masked randomly. The results indicate that MiniTraffic exhibits excellent reconstruction performance on non-masked patches and can effectively model masked patches. We extend this reconstruction ability to the imputation task. **Figures 4(c)** and **4(d)** show that when faced with missing values, pre-trained MiniTraffic also excels in reconstructing the missing patches. This demonstrates that MiniTraffic not only performs well during the pre-training phase but also effectively applies its reconstruction ability to practical imputation tasks.

## 5.7. Long-horizon Prediction

While lane-level traffic is dominated by short-term variability, we further evaluate MiniTraffic at extended horizons of 24 and 48 steps on PeMS-Lane and HuaNan-Lane to assess its long-range capability. As shown in **Figure 5**, prediction errors increase monotonically and smoothly with the horizon, without divergence or collapse, indicating that the semantic-driven sparse graph and patch-level representation preserve enough long-range structure for stable medium-to-long-horizon forecasting.

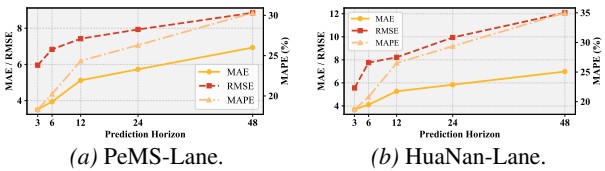

| *(a)* PeMS-Lane. | *(b)* HuaNan-Lane. |

*Figure 5.* Long-horizon prediction results at different horizons.

## 5.8. Cross-city Transfer

To probe inter-city generalization, we pre-train MiniTraffic on HuaNan and fine-tune on PeMS-Lane under two settings: same-granularity (HuaNan-Lane $\rightarrow$ PeMS-Lane) and cross-granularity (HuaNan-Road $\rightarrow$ PeMS-Lane). As reported in **Table 6**, both transfer settings incur higher errors than the within-city baseline due to distribution shifts in traffic patterns and spectral structures, but the same-granularity transfer consistently outperforms the cross-granularity one across all horizons. This indicates that granularity alignment between source and target domains is more important than co-location for effective transfer.

*Table 6.* Cross-city and cross-domain transfer performance on PeMS-Lane.

| Transfer | HuaNan-Lane $\rightarrow$ PeMS-Lane | | | HuaNan-Road $\rightarrow$ PeMS-Lane | | |
|---|---|---|---|---|---|---|
| Horizon | MAE | RMSE | MAPE | MAE | RMSE | MAPE |
| 3 | 4.53 | 7.48 | 24.65% | 4.90 | 8.08 | 26.61% |
| 6 | 5.08 | 8.57 | 27.24% | 5.48 | 9.26 | 29.40% |
| 12 | 6.59 | 9.32 | 32.75% | 7.13 | 10.07 | 35.36% |

## 5.9. Robustness of FDA

The frequency-domain perturbations in FDA are bounded by two principled constraints: an amplitude threshold $\epsilon(f) = \lambda \cdot \max A(f)$ with $\lambda \in (0, 1)$, and a selective spectral mask $\Gamma(f)$ that confines perturbation to dominant frequency bands. Under these constraints, the relative spectral energy shift is bounded by $\lambda^2 \cdot \sum_f \Gamma(f) A(f)^2 / \sum_f A(f)^2$, ensuring that the global signal structure is preserved while controlled variability is injected. Empirically, the model degrades gracefully when $\lambda$ deviates from its intended range and remains stable across datasets, confirming that FDA-augmented samples stay within a Lipschitz-bounded neighborhood of the original signal manifold rather than introducing label noise. Full theoretical derivations and sensitivity studies are provided in **Appendix E**.

# 6. Conclusion and Future Work

We frame the unified prediction of roads and lanes as a fine-grained traffic prediction problem and introduce **Mini-Traffic**, the first lightweight model specifically designed for this task via road-level pre-training. MiniTraffic employs efficient strategies such as Frequency Domain Stability Augmentation and Contrastive Clustering Graph Partitioning to leverage abundant road-level data and mitigate the scarcity of lane-level annotations. These mechanisms enhance transferability while significantly reducing parameter overhead, enabling deployment in resource-constrained scenarios. Through granularity-aware fine-tuning, the model supports efficient multi-level adaptation with low training cost. Future work may explore applying this framework to broader downstream tasks such as lane-change intention prediction and traffic signal optimization, as well as integrating fine-grained forecasting into large-scale foundation models to enhance their spatio-temporal reasoning capabilities.

## Acknowledgements

This work was supported by the China Scholarship Council (Grant No. 202506100105) and the Youth Talent Support Project by the China Association for Science and Technology. The computations in this research were performed on the CFFF platform of Fudan University. We also thank the anonymous reviewers and Area Chairs for their constructive comments and support.

## Impact Statement

This paper introduces a lightweight pre-training framework, MiniTraffic, designed to enhance fine-grained traffic tasks. By improving the efficiency and scalability of traffic models, our work has the potential to contribute to smarter urban traffic management and reduce environmental impacts by optimizing traffic flow. We foresee no immediate ethical concerns or societal risks associated with this research.

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

## A. Impact of Fine-tuning Data Ratio

To assess the few-shot generalization capability of MiniTraffic, we conduct fine-tuning experiments using different proportions of lane-level data—10%, 30%, and 60%—with a fixed prediction horizon of $h = 6$. **Figure 6** shows the model's performance across three benchmark datasets: PeMS-Lane, HuaNan-Lane, and PeMSF-Lane.

Across all datasets, MiniTraffic demonstrates a consistent performance improvement as the amount of fine-tuning data increases. Importantly, the model achieves reasonable accuracy even when only 10% of labeled data is available, underscoring its ability to effectively transfer knowledge from road-level pre-training to lane-level prediction.

Specifically, on the PeMS-Lane dataset, the MAE decreases from 4.75 at 10% data to 3.94 at 60%; on HuaNan-Lane, from 4.93 to 4.11; and on PeMSF-Lane, from 4.98 to 4.15. This robustness under low-data regimes confirms MiniTraffic's practicality for real-world applications where lane-level annotations are scarce. The results validate the framework's few-shot adaptability and its effectiveness for fine-grained traffic prediction with minimal supervision.

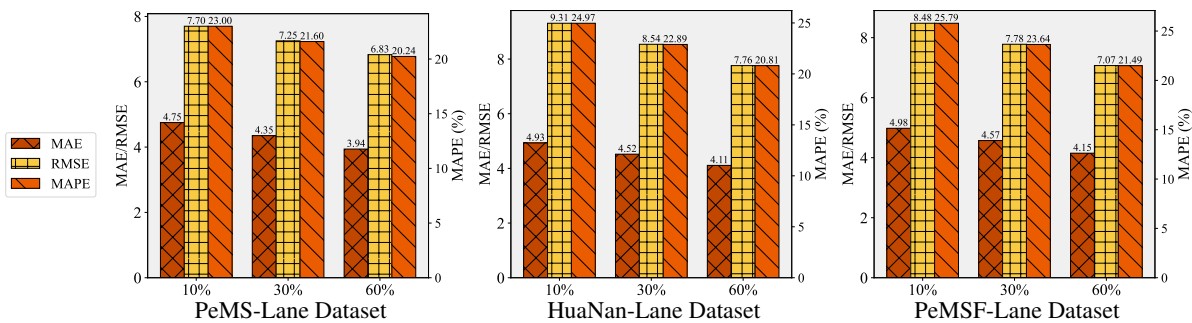

*Figure 6.* Performance comparison under different data ratios on $h = 6$ for three datasets.

## B. Cost Analysis

### B.1. Training and Inference Cost Analysis

One of the core design goals of MiniTraffic is to remain lightweight with a small parameter scale, making it adaptable to resource-constrained scenarios for multi-granularity traffic prediction tasks. To further validate this objective, we conducted additional comparisons of training and inference costs on the PeMS-(Road/Lane) datasets (horizon=6), as shown in **Figure 7**. It is important to clarify that for models involving a pre-training phase (MiniTraffic, GPT-ST, FlashST), the reported "Training Cost" is measured during pre-training, while "Inference Cost" is measured during the fine-tuning phase. FLOPs are calculated under the same principle: for lane-level tasks, FLOPs are measured in the fine-tuning stage, while for road-level tasks, they are measured in the pre-training stage.

The results show that MiniTraffic significantly reduces both training and inference overhead while maintaining predictive accuracy. In lane-level tasks, MiniTraffic's FLOPs during fine-tuning are substantially reduced compared to pre-training, and its inference latency is lower than most task-specific models, demonstrating high efficiency in small-scale and dynamic scenarios. In road-level tasks, MiniTraffic achieves much lower training and inference costs than large-scale pre-training models such as GPT-ST, with inference latency reduced by over 40% and FLOPs reduced by about 85%. These findings confirm that the frequency-domain augmentation and contrastive clustering modules not only improve predictive performance but also make MiniTraffic highly practical for deployment in resource-limited environments.

### B.2. Fine-tuning Cost Analysis

To further demonstrate the lightweight nature of MiniTraffic, we report both the parameter size (in thousands) and the iteration-level fine-tuning time (in $10^{-2}$ seconds) under the prediction horizon $h = 6$. **Table 7** summarizes the results across road- and lane-level tasks for the PeMS-(Road/Lane), HuaNan-(Road/Lane), and PeMSF-(Road/Lane) datasets. MiniTraffic demonstrates consistently low parameter overhead and rapid fine-tuning convergence across all datasets. Notably, even the lane-level variants—which are inherently more complex—require no more than 16.3k parameters and under $2 \times 10^{-2}$ seconds per iteration for fine-tuning.

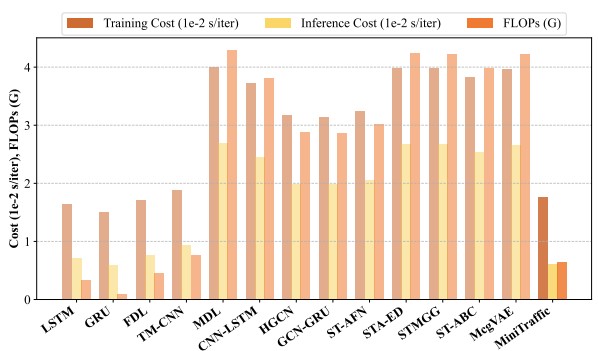

*(a)* Training and Inference Costs of Lane-Level Task Models

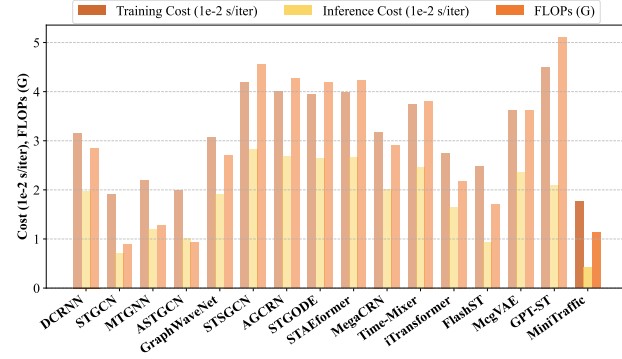

*(b)* Training and Inference Costs of Road-Level Task Models

*Figure 7.* Comparison of Training and Inference Costs between Task-Specific Models and MiniTraffic.

This compact design is particularly advantageous for real-world deployment, especially on edge devices where memory and computational resources are limited. Beyond edge scenarios, such efficiency enables broader scalability, reduced energy consumption, and faster adaptation in dynamic traffic environments, reinforcing MiniTraffic's practicality in diverse applications.

*Table 7.* Parameter size and fine-tuning time per iteration at $h = 6$.

| Datasets | PeMS | | HuaNan | | PeMSF | |
|---|---|---|---|---|---|---|
| Task | Road | Lane | Road | Lane | Road | Lane |
| Para (k) | 1.8 | 9.1 | 4.1 | 16.3 | 1.8 | 9.8 |
| Cost ($10^{-2}$s/itr) | 0.5 | 1.6 | 1.1 | 1.9 | 0.5 | 1.7 |

### B.3. Scalability on Large-Scale Road Networks

Although our lane-level experiments mainly target regional scenarios (e.g., signal control and tidal-lane management), it is also important to assess whether MiniTraffic can be reliably trained on large-scale road networks and to quantify its computational cost. To this end, we pre-train MiniTraffic on a city-scale dataset, the GBA subset of LargeST (Liu et al., 2023b), which contains 2,352 sensors, and log the corresponding training resource usage. On the LargeST-GBA dataset, the measured training cost is summarized in **Table 8**.

These results show that, even when the number of nodes increases substantially, MiniTraffic can still be trained with low per-iteration time and a modest parameter count, which is consistent with our design goal of being both lightweight and scalable.

*Table 8.* Training cost on the GBA dataset.

| Training Cost (1e-2 s/iter) | FLOPs (G) | Param (k) |
|---|---|---|
| 4.47 | 4.51 | 237k |

## C. Analysis of Trainable Parameter vs. Performance

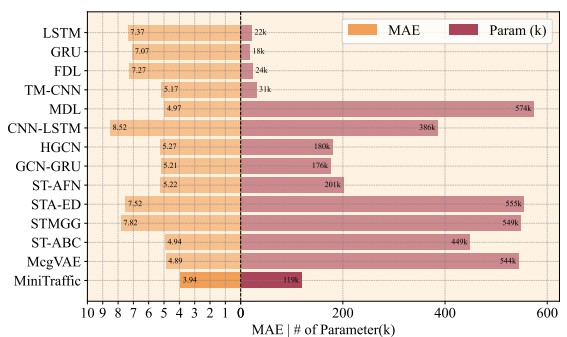
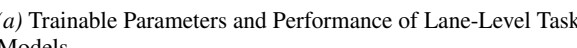

*(a)* Trainable Parameters and Performance of Lane-Level Task Models

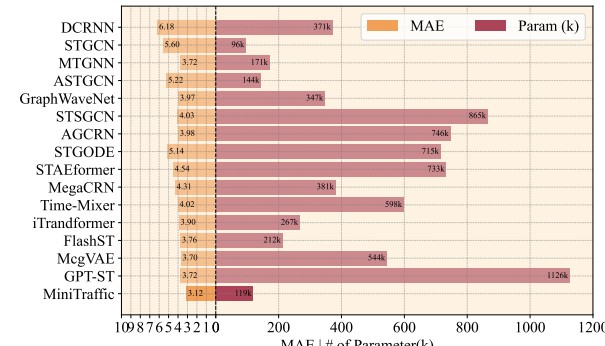

*(b)* Trainable Parameters and Performance of Road-Level Task Models

*Figure 8.* Comparison of Trainable Parameters between Task-Specific Models and MiniTraffic.

We conducted a comparative analysis of the performance of task-specific models and pre-trained models (FlashST, GPT-ST and MiniTraffic) under the main experimental conditions, focusing on their trainable parameter count and prediction error (MAE) on the PeMS-(Road/Lane) datasets (horizon=6). It is important to note that comparing parameter counts during the

fine-tuning phase is not entirely fair for task-specific models; therefore, we emphasize the parameter scale and performance during the pre-training phase. As shown in **Figures 8**, MiniTraffic is referred to as a "mini pre-trained model" because its trainable parameter count during the pre-training phase is only 119k, even lower than most task-specific models. In comparison, only shallow neural networks (such as LSTM and GRU) and the fully convolutional STGCN have fewer parameters, yet their prediction performance is significantly inferior to MiniTraffic. This demonstrates that MiniTraffic effectively reduces the computational overhead of traditional graph attention mechanisms and minimizes model size through contrastive clustering and fine-grained graph attention. Additionally, this design enables the establishment of correlations between road and lane information, facilitating efficient modeling of multi-granularity traffic data.

This design not only significantly reduces computational resource requirements but also ensures efficient performance in resource- and data-constrained environments, fully showcasing MiniTraffic's strong advantages in practical traffic prediction tasks.

## D. Lane-to-Road Relationship and Future Discussion

We observe a natural compositional relationship between lane-level and road-level traffic: a road is formed by its constituent lanes. This raises a natural question: can we simply average the predicted future speeds of all lanes on a road and treat this as the road's future speed? To explore this, we use MiniTraffic to obtain lane-level predictions with a horizon of 6 on three datasets, aggregate the corresponding $J$ lanes by averaging, and compare the resulting series with the road-level ground truth; the errors are reported in **Table 9**. Across all three datasets, the Lanes

*Table 9.* Comparison between the Lane (Mean) for Road and direct road-level prediction.

| Tasks | Lane(Mean) for Road | | | Road | | |
|---|---|---|---|---|---|---|
| Datasets | MAE | RMSE | MAPE | MAE | RMSE | MAPE |
| PeMS | 3.81 | 6.19 | 15.74% | 3.12 | 4.90 | 11.22% |
| HuaNan | 3.90 | 6.16 | 16.90% | 3.16 | 4.46 | 12.45% |
| PeMSF | 3.91 | 6.52 | 16.60% | 3.17 | 5.21 | 11.76% |

(Mean) for Road baseline consistently yields higher errors than direct road-level modeling. This suggests that naive averaging introduces error accumulation from individual lanes and requires much stricter spatio-temporal alignment and imputation between lanes and roads, effectively demanding high accuracy on every lane. A unified multi-granularity framework that first achieves accurate lane-level modeling and then reconstructs road-level states is therefore a promising research direction, but under the current setting, simple averaging is still insufficient to replace dedicated road-level prediction.

## E. Supplementary Analysis of the FDA Module

In the MiniTraffic model, the FDA module plays a key role by applying frequency domain perturbations to temporal data, simulating multi-lane traffic characteristics, and enhancing the model's performance on fine-grained traffic prediction tasks. The following analysis delves into the working principles of the FDA module, exploring aspects such as frequency domain transformation, noise constraints, and the impact of perturbations on model performance. We also provide mathematical derivations and theoretical proofs to clarify how this module contributes to improving the model's generalization capability.

### E.1. Frequency Domain Transformation and Its Relation to Temporal Features

The FDA module first projects road-level traffic signals from the time domain into the frequency domain using the DFT. We adopt a matrix form of DFT for efficient computation and compatibility with batched traffic data:

$$\tilde{X}^R = X^R \cdot F_T, \quad \text{where } [F_T]_{t,f} = e^{-j\frac{2\pi(t-1)(f-1)}{T}}, \quad X^R \in \mathbb{R}^{N^R \times T}. \tag{20}$$

Each transformed sequence $\tilde{X}^R(f)$ can be decomposed as $A(f)e^{j\theta(f)}$, where $A(f)$ captures the power spectrum and $\theta(f)$ encodes phase alignment. This representation allows the model to analyze temporal patterns from a frequency-centric perspective:

- **Low-frequency components** ($f \approx 0$) retain long-term trends and stable flows.

- **High-frequency components** are sensitive to short-term fluctuations and localized dynamics (e.g., traffic bursts, signal switching).

Importantly, perturbing selective frequency bands allows us to inject diversity without altering global trends. For example, modifying only high-frequency magnitude minimally changes the long-range mean pattern in the original time domain. The Fourier basis ensures orthogonality, so localized changes do not diffuse across components—this is essential for preserving structural consistency while introducing variation.

### E.2. Simulating Multi-Lane Data Variability: Frequency Domain Perturbations

Once the signal is mapped into the frequency domain, the FDA module introduces structured perturbations to mimic lane-level variability. This is grounded in the observation that while different lanes may follow similar global patterns (e.g., overall traffic trend), they exhibit subtle differences in local intensity and phase.

To capture such fine-grained diversity, we model perturbations on both the magnitude and phase of each frequency component:

$$\tilde{X}^R(f) = (A(f) + \delta_A(f)) \cdot e^{j(\theta(f) + \delta_\theta(f))}, \tag{21}$$

where $\delta_A(f) \sim \mathcal{N}(0, \sigma_A^2(f))$, $\delta_\theta(f) \sim \mathcal{N}(0, \sigma_\theta^2(f))$ introduce stochastic diversity.

However, not all frequencies are equally informative or safe to perturb. We introduce two principled constraints:

**(1) Amplitude constraint:** $|\delta_A(f)| \leq \epsilon(f) = \lambda \cdot \max A(f)$ ensures that the perturbation magnitude is proportional to signal strength and bounded by a controllable hyperparameter $\lambda \in (0, 1)$.

**(2) Selective masking:** We construct a binary frequency mask $\Gamma(f) = \mathbb{I}(A(f)^2 > \tau \cdot \max A(f)^2)$ to restrict perturbation to the most informative bands. Here, $\tau$ is a learnable parameter that adapts to dataset-specific spectral patterns.

The resulting constrained perturbation becomes:

$$\tilde{X}^R(f) = (A(f) + \Gamma(f) \cdot \delta_A(f)) \cdot e^{j(\theta(f) + \Gamma(f) \cdot \delta_\theta(f))}. \tag{22}$$

This design offers several advantages:

**Lane-awareness:** Different frequency bands encode different lane-scale dynamics; perturbing them independently can simulate real lane divergence.

**Lane Transferability:** Since road-level data is abundant and lane-level data scarce, this frequency-level augmentation creates pseudo-lane variations without needing real lane annotations.

**Lane Controllability:** The explicit formulation allows systematic control over the extent of augmentation, balancing between stability and expressiveness.

To ensure stability, we analyze the relative energy shift between perturbed and original signals:

$$\left| \frac{\|\tilde{X}^R(f)\|_2^2 - \|X^R(f)\|_2^2}{\|X^R(f)\|_2^2} \right| \lesssim \lambda^2 \cdot \frac{\sum_f \Gamma(f) A(f)^2}{\sum_f A(f)^2}. \tag{23}$$

This bound, derived from Parseval's identity and the perturbation structure, ensures that global spectral energy remains nearly invariant.

Finally, the signal is reconstructed into the time domain using the inverse DFT:

$$\tilde{X}^R = \Re(\tilde{X}^R(f) \cdot F_T^{-1}). \tag{24}$$

This real-valued output retains the original temporal format while embedding frequency-induced variations that emulate lane-level dynamics.

### E.3. Properties and Empirical Impact of Noise Constraints

The frequency-domain perturbation strategy is regulated by two core constraints:

- **Amplitude Constraint:** $|\delta_A(f)| \leq \epsilon(f) = \lambda \cdot \max A(f)$, with $\lambda \in (0, 1)$.

- **Frequency Masking:** $\Gamma(f) = \mathbb{I}(A(f)^2 > \tau \cdot \max A(f)^2)$, with $\tau$ learnable.

These constraints prevent over-distortion of the input while still introducing structured variation. The relative energy stability constraint ensures that the global shape of the signal remains intact, even under perturbation.

**Theoretical Motivation.** To further support our design, we draw upon the generalization error bound from statistical learning theory:

$$\mathbb{E}_{\text{test}}[L(f)] \leq \mathbb{E}_{\text{train}}[L(f)] + \mathcal{O}\left(\frac{1}{\sqrt{D}}\right), \tag{25}$$

where $D$ is data diversity. Moderate perturbations in the frequency domain increase diversity and improve generalization. However, overly strong noise—i.e., large $\epsilon$—can break the structural correspondence between road- and lane-level signals, reducing data fidelity and increasing training error. This trade-off motivates our introduction of task-specific amplitude and frequency constraints.

**Role of $\lambda$.** The amplitude threshold $\epsilon(f)$ is dynamically scaled using a coefficient $\lambda$, defined as:

$$\epsilon(f) = \lambda \cdot \max A(f). \tag{26}$$

To assess the sensitivity of MiniTraffic to the choice of $\lambda$, we conducted lane-level experiments using two ranges: $\lambda \in (0, 1)$ and $\lambda \in (1, 2)$. This empirically confirms the importance of keeping noise strength within a bounded and meaningful interval. As shown in **Table 10**, exceeding the ideal perturbation strength leads to performance drop, especially in RMSE and MAPE. We also observe that HuaNan-Lane is the most sensitive to larger $\lambda$, while PeMS-Lane and PeMSF-Lane exhibit similar but slightly milder degradation, indicating that FDA is robust across datasets as long as $\lambda$ stays within a moderate range. These results validate our theoretical assumption and support the practical choice of $\lambda \in (0, 1)$ for stable and effective augmentation in fine-grained traffic prediction.

*Table 10.* Effect of $\lambda$ on lane-level prediction performance.

| $\lambda$ | $\lambda \in (0, 1)$ | | | $\lambda \in (1, 2)$ | | |
|---|---|---|---|---|---|---|
| **Metrics** | MAE | RMSE | MAPE | MAE | RMSE | MAPE |
| PeMS-Lane | 3.98 | 6.90 | 20.44% | 4.91 | 9.11 | 25.02% |
| HuaNan-Lane | 4.10 | 7.75 | 20.63% | 5.42 | 10.23 | 27.23% |
| PeMSF-Lane | 4.16 | 7.34 | 21.88% | 5.19 | 9.29 | 26.88% |

### E.4. Impact of Perturbations on Model Generalization Performance

Controlled perturbations serve as a form of implicit data regularization. From the perspective of statistical learning theory, introducing label-preserving variations improves the expected generalization performance by increasing the effective coverage of the hypothesis space.

Formally, let $\mathcal{F}$ be the model class, and $\mathcal{R}_n(\mathcal{F})$ its empirical Rademacher complexity. Then the expected generalization error admits the upper bound:

$$\mathbb{E}_{\text{test}}[L(f)] \leq \mathbb{E}_{\text{train}}[L(f)] + \mathcal{R}_n(\mathcal{F}) + \mathcal{O}\left(\frac{1}{\sqrt{n}}\right). \tag{27}$$

The FDA module expands the training support through structured augmentation, effectively increasing the number of distinct yet semantically consistent instances. In turn, this reduces the empirical complexity $\mathcal{R}_n(\mathcal{F})$, leading to tighter generalization bounds.

In our case, frequency domain perturbations introduce fine-grained, low-energy distortions aligned with the signal manifold, enabling better coverage of the variation space without label noise. This aligns with the intuition that $\mathcal{E} \propto 1/\sqrt{\mathcal{D}}$, where $\mathcal{D}$ denotes the effective sample diversity.

Moreover, since the perturbation is bounded (via $\lambda$) and targeted (via $\Gamma(f)$), the augmented samples remain within a Lipschitz-bounded neighborhood of the original signal manifold. This local consistency ensures that the model does not overfit to synthetic artifacts but instead generalizes through smooth expansion of the learning domain.

## E.5. Impact of Noise Constraints on the Effectiveness of the Generated Data

The amplitude constraint $\lambda$ and frequency mask threshold $\tau$ jointly regulate the perturbation's strength and locality. These parameters directly influence the trade-off between expressiveness and stability.

When $\lambda$ is too large, the magnitude perturbation $\delta_A(f)$ can alter the signal's spectral identity, pushing the augmented sample outside the smooth generalization region. Conversely, too small a $\lambda$ yields negligible diversity. Thus, moderate settings ensure that perturbations expand the hypothesis support without causing label inconsistency or signal collapse.

Similarly, the frequency mask $\Gamma(f)$ adapts the augmentation scope based on energy concentration. Too low a threshold $\tau$ over-expands $\Gamma(f)$, introducing random high-frequency noise. A well-calibrated $\tau$ focuses perturbation on dominant harmonics, enhancing semantically aligned variation.

Together, $(\lambda, \tau)$ serve as spectral-domain analogues of spatial augmentation strength and region selection in image tasks, offering a tunable balance between model regularization and input fidelity.

# F. Theoretical Expressive Power of Small-Scale Graph Structures

In the research of graph neural networks, the expressive power of a model is often closely related to its flexibility and accuracy in handling graph structures. While global graph structures can fully capture all relationships between nodes, their high time complexity and memory requirements pose limitations for practical applications. To address this issue, MiniTraffic introduces small-scale graph structures by partitioning the global graph into several subgraphs using contrastive clustering, significantly reducing computational cost. However, this raises a fundamental question: can small-scale graphs approximate or achieve equivalence to global graph structures in terms of node relationship representation? This section discusses the expressive power of small-scale graphs through theoretical analysis and mathematical derivation, examining both graph representational capacity and computational complexity, verifying that small-scale graphs maintain expressive capability while improving computational efficiency.

## F.1. Analysis of Graph Representational Capacity

Small-scale graph structures decompose the global graph into several smaller graphs. By using contrastive clustering to group similar nodes, they build local adjacency matrices. Whether this design is equivalent to or closely approximates the global graph structure in terms of node relationship representation can be analyzed through the following derivation.

Let the global graph's node set be denoted as $\mathcal{V}$, edge set as $\mathcal{E}$, and adjacency matrix as $\mathbf{A} \in \mathbb{R}^{N \times N}$, where $N = |\mathcal{V}|$. Small-scale graphs use contrastive clustering to group similar nodes, with each group forming a subgraph $\mathcal{V}_i$, whose adjacency matrix is $\mathbf{A}_i \in \mathbb{R}^{|\mathcal{V}_i| \times |\mathcal{V}_i|}$, satisfying the following property:

$$\mathbf{A}_i[u,v] = \mathbf{A}[u,v], \quad \forall u, v \in \mathcal{V}_i. \tag{28}$$

The expressive power of small-scale graph structures can be analyzed by examining the eigenvalues of their Laplacian matrices. The normalized Laplacian matrix of the global graph is:

$$\mathbf{L} = \mathbf{I} - \mathbf{D}^{-1/2}\mathbf{A}\mathbf{D}^{-1/2}, \tag{29}$$

where $\mathbf{D}$ is the degree matrix. For subgraph $\mathcal{V}_i$, its Laplacian matrix is:

$$\mathbf{L}_i = \mathbf{I}_i - \mathbf{D}_i^{-1/2}\mathbf{A}_i\mathbf{D}_i^{-1/2}. \tag{30}$$

If the edge weights between subgraphs are small, i.e., the boundary nodes generated by contrastive clustering are highly independent, the spectral properties of the global graph can be approximated by the spectral properties of the subgraphs. By spectral decomposition, let $\mathbf{L}$ have eigenvalues $\{\lambda_1, \lambda_2, \ldots, \lambda_N\}$ and $\mathbf{L}_i$ have eigenvalues $\{\lambda_1^i, \lambda_2^i, \ldots, \lambda_{|\mathcal{V}_i|}^i\}$, we have:

$$\bigcup_i \{\lambda_1^i, \lambda_2^i, \ldots, \lambda_{|\mathcal{V}_i|}^i\} \approx \{\lambda_1, \lambda_2, \ldots, \lambda_N\}. \tag{31}$$

Thus, the spectral expressive power of small-scale graphs approximates that of the global graph, ensuring the model's representational capability.

### F.2. Theoretical Trade-off Between Computational Complexity and Effectiveness

By comparing the computational complexity of global graphs and small-scale graphs, we can quantify the reduction in time complexity and potential precision loss.

**Computational Complexity of Global Graph:** When applying graph attention mechanisms to the global graph, the time complexity is:

$$\mathcal{O}(N^2 d), \tag{32}$$

where $d$ is the feature dimension.

**Computational Complexity of Small-Scale Graph:** For each subgraph $\mathcal{V}_i$, suppose the number of nodes in the subgraph is $n_i$. The time complexity of the small-scale graph is:

$$\mathcal{O}\left(\sum_{i=1}^{k} n_i^2 d\right), \tag{33}$$

where $k$ is the number of subgraphs, satisfying $\sum_{i=1}^{k} n_i = N$. When each subgraph is small, $n_i \ll N$, the computational complexity is significantly reduced.

By optimizing the subgraph partition through clustering methods, ensuring that each $n_i$ is close to the average $\frac{N}{k}$, the computational complexity further simplifies to:

$$\mathcal{O}\left(k \cdot \left(\frac{N}{k}\right)^2 \cdot d\right) = \mathcal{O}\left(\frac{N^2 d}{k}\right), \tag{34}$$

which shows a $k$-fold reduction in time complexity compared to the global graph.

### F.3. Practical Efficiency and Precision Loss

In practical tasks, precision loss can be quantified by the difference between the adjacency matrix of the global graph $\mathbf{A}$ and that of the small-scale graph $\mathbf{A}^*$:

$$\Delta = \|\mathbf{A} - \mathbf{A}^*\|_F, \tag{35}$$

where $\|\cdot\|_F$ denotes the Frobenius norm. Experimental validation shows that $\Delta$ is primarily determined by the similarity measure of contrastive clustering and the choice of $k$. With an appropriate choice of $k$ and clustering strategy, $\Delta$ can be maintained within a small range, thus significantly reducing computational complexity while preserving the model's predictive accuracy.

In summary, the small-scale graph structure effectively reduces computational complexity through local graph construction. Theoretical analysis shows that its expressive power in the spectral domain approximates that of the global graph, and experimental results verify that it achieves a good balance between computational efficiency and precision loss.

## G.  Complexity Analysis

MiniTraffic, as a lightweight pre-trained model, employs various strategies in its design to minimize both time and space complexity.

### G.1. Time Complexity Analysis

**1. Pre-training Stage:**

a. Frequency Domain Stability Augmentation (FDA): The FDA is implemented via the Fast Fourier Transform (FFT), with a time complexity of $O(T \log T)$, where $T$ is the length of the time series. This reduces the computational load compared to traditional models.

b. Contrastive Clustering and Adjacency Matrix Generation: For $k$ small-scale subgraphs with feature dimension $d$, calculating cosine similarities has a time complexity of $O(k^2 d)$. However, by retaining only the most similar $m$ neighbors for each subgraph, the storage and computation of the adjacency matrix are significantly reduced.

c. Graph Attention Convolution: The time complexity for graph attention is $O(N^2 d)$, where $N$ is the number of nodes in the global graph. In MiniTraffic, each small-scale graph has fewer nodes ($n_i$), reducing the time complexity to $O(n_i^2 d)$, thus cutting down computational costs.

**2. Fine-tuning Stage:** Fine-tuning only updates specific parameters (e.g., Adaptive Head, Reduction Head), reducing the time complexity to $O(T)$, where $T$ is the size of the training dataset. This significantly shortens training time compared to full training.

**G.2. Space Complexity Analysis**

**1. Model Parameters:** MiniTraffic has only 119k parameters, much lower than many other pre-trained models, leading to reduced memory and GPU usage.

**2. Sparse Adjacency Matrix:** Unlike traditional dense adjacency matrices, MiniTraffic uses sparse matrices generated via contrastive clustering. The space complexity of the adjacency matrix is reduced from $N^2$ (in traditional models) to $n_i^2$, further optimized by retaining only the most similar neighbors.

**3. Graph Convolution Operation:** MiniTraffic uses sparse graph structures, reducing space complexity from $O(N^2)$ (in traditional models) to $O(n_i^2)$, significantly decreasing storage and computation needs.

By leveraging fine-grained graph attention, efficient contrastive clustering, and a pre-training-fine-tuning strategy, MiniTraffic reduces both computational and memory requirements while maintaining strong prediction performance. Its lightweight design makes it ideal for fine-grained traffic prediction tasks, especially in resource-constrained environments.

# H. Dataset Description

In this section, we provide additional details regarding the datasets used for pre-training and fine-tuning tasks. **Tables 11** and **Tables 12** present the statistical information for the pre-training and fine-tuning datasets, respectively.

**METR-LA (Road):** This traffic dataset contains information collected from loop detectors on highways in Los Angeles County. We selected 207 sensors and gathered four months of data from March 1, 2012, to June 30, 2012, totaling 6,519,002 observed traffic data points.

**PeMS-Bay (Road):** This dataset was collected by the California Department of Transportation's (CalTrans) Performance Measurement System (PeMS). We selected 325 sensors in the Bay Area and collected six months of data from January 1, 2017, to May 31, 2017, with a total of 16,937,179 observed traffic data points.

**PeMS-(Road/Lane):** This dataset originates from the Santa Ana Freeway in Los Angeles, USA, and includes data from 8 sensors. As part of the PeMS public dataset, it includes traffic speed information from February 5 to March 5, 2017, with each road segment covering data from five regular lanes. The dataset comprises 64,472 observed road-level traffic data points and 322,360 lane-level traffic data points.

**PeMSF-(Road/Lane):** This dataset also comes from the Santa Ana Freeway in Los Angeles, USA, and includes data from 8 sensors. As part of the PeMS public dataset, it covers traffic speed information from February 5 to March 5, 2017. Unlike the PeMS dataset, each road segment in this dataset covers either five or six irregular lanes. The dataset includes 64,472 observed road-level traffic data points and 346,537 lane-level traffic data points.

**HuaNan-(Road/Lane):** This dataset was collected from the Huanan Expressway in Guangzhou, China, and includes data from 18 sensors. The data collection period spans 30 days, from July 22 to August 22, 2022. This dataset is representative of urban road traffic patterns, with 803,520 observed road-level traffic data points and 3,214,080 lane-level traffic data points.

Among these, METR-LA and PeMS-Bay are the most commonly used benchmark datasets for road-level traffic prediction (Li et al., 2023a)[3], while PeMS-(Road/Lane), PeMSF-(Road/Lane), and HuaNan-(Road/Lane) are the most commonly used benchmark datasets for fine-grained traffic prediction (Li et al., 2025a)[4].

---

[3]https://github.com/tsinghua-fib-lab/Traffic-Benchmark
[4]https://github.com/ShuhaoLii/LaneLevel-Traffic-Benchmark

*Table 11.* Statistics of Pre-training Datasets.

| Dataset | Timespan | # of Objects | Unit | Interval |
|---------|----------|--------------|------|----------|
| METR-LA | 3/1/2012-6/30/2012 | 6,519,002 | Miles/Hour | 5 min |
| PeMS-Bay | 1/1/2017-5/31/2017 | 16,937,179 | Miles/Hour | 5 min |
| PeMS-Road | 2/5/2017-3/5/2017 | 64,472 | Miles/Hour | 5 min |
| PeMSF-Road | 2/5/2017-3/5/2017 | 64,472 | Miles/Hour | 5 min |
| HuaNan-Road | 7/22/2022-8/22/2022 | 803,520 | Kilometers /Hour | 2 min |

*Table 12.* Statistics of Fine-tuning Datasets.

| Dataset | Timespan | # of Objects | Unit | Interval |
|---------|----------|--------------|------|----------|
| PeMS-Road | 2/5/2017-3/5/2017 | 64,472 | Miles/Hour | 5 min |
| PeMSF-Road | 2/5/2017-3/5/2017 | 64,472 | Miles/Hour | 5 min |
| HuaNan-Road | 7/22/2022-8/22/2022 | 803,520 | Kilometers /Hour | 2min |
| PeMS-Lane | 2/5/2017-3/5/2017 | 322,360 | Miles/Hour | 5 min |
| PeMSF-Lane | 2/5/2017-3/5/2017 | 346,537 | Miles/Hour | 5 min |
| HuaNan-Lane | 7/22/2022-8/22/2022 | 3,214,080 | Kilometers /Hour | 2 min |

## I. Evaluation Metrics Formulas

To ensure a fair assessment of prediction performance, we employed three commonly used evaluation metrics in traffic prediction: Mean Absolute Error (MAE), Root Mean Square Error (RMSE), and Mean Absolute Percentage Error (MAPE). MAE and RMSE are used to measure absolute prediction errors, while MAPE is used to measure relative prediction errors.

**Formula for Road-Level Tasks:**

$$MAE = \frac{1}{h} \sum_{t=1}^{h} \frac{1}{N^R} \sum_{i=1}^{N^R} \left| y_t^{r_i} - \hat{y}_t^{r_i} \right|, \tag{36}$$

$$RMSE = \sqrt{\frac{1}{h} \sum_{t=1}^{h} \frac{1}{N^R} \sum_{i=1}^{N^R} \left( y_t^{r_i} - \hat{y}_t^{r_i} \right)^2}, \tag{37}$$

$$MAPE = \frac{1}{h} \sum_{t=1}^{h} \frac{1}{N^R} \sum_{i=1}^{N^R} \left| \frac{y_t^{r_i} - \hat{y}_t^{r_i}}{y_t^{r_i}} \right|. \tag{38}$$

**Formula for Lane-Level Tasks:**

$$MAE = \frac{1}{h} \sum_{t=1}^{h} \frac{1}{N^L} \sum_{i=1}^{I} \sum_{j=1}^{J_i} \left| y_t^{l_{i,j}} - \hat{y}_t^{l_{i,j}} \right|, \tag{39}$$

$$RMSE = \sqrt{\frac{1}{h} \sum_{t=1}^{h} \frac{1}{N^L} \sum_{i=1}^{I} \sum_{j=1}^{J_i} \left( y_t^{l_{i,j}} - \hat{y}_t^{l_{i,j}} \right)^2}, \tag{40}$$

$$MAPE = \frac{1}{h} \sum_{t=1}^{h} \frac{1}{N^L} \sum_{j=1}^{J_i} \sum_{i=1}^{I} \sum_{j=1}^{J_i} \left| \frac{y_t^{l_{i,j}} - \hat{y}_t^{l_{i,j}}}{y_t^{l_{i,j}}} \right|. \tag{41}$$

## J. Baseline Details

In this section, we detail the **29** baseline models employed for various fine-grained traffic prediction tasks and describe their implementation specifics.

## J.1. Pre-trained Models

**FlashST** (Li et al., 2024d): FlashST is a lightweight prompt-tuning framework that adapts pre-trained spatio-temporal models to diverse downstream traffic datasets via in-context learning and distribution alignment. While it achieves strong generalization across domains, its reliance on prompt design may limit adaptability to highly dynamic traffic patterns. We adopt the official implementation[5] and follow the same pretraining settings as MiniTraffic for fair comparison, using the best-performing MTGNN variant described in the original paper as the base model.

**GPT-ST** (Li et al., 2023c) : GPT-ST combines generative pretraining with spatio-temporal graph neural networks, effectively capturing global spatio-temporal dependencies to improve the generalization capability of sequence prediction tasks. In our experiments, we adopted Graph WaveNet, which demonstrated the best performance in the original paper, as the base model and trained it using the same pretraining datasets as MiniTraffic. The implementation is based on the official code provided by the authors[6].

## J.2. Multi-task Models

**McgVAE** (Li et al., 2024b): McgVAE is a multi-channel graph-structured variational autoencoder that integrates road-level information to provide a global perspective for lane-level prediction. It performs comprehensive tasks through three interconnected channels, each capturing spatio-temporal information at different granularities. Notably, McgVAE requires both road-level and corresponding lane-level data as input. The implementation in our experiments is based on the official code released by the authors[7].

## J.3. Lane-Level Task Models

**Cat-RF-LSTM** (Zhao & Chen, 2022): This hybrid model combines CatBoost for constructing spatio-temporal features, Random Forest for variance reduction, and LSTM for extracting temporal trends. The final prediction is made using a stacking ensemble approach. The model is implemented using CatBoost, scikit-learn, and PyTorch libraries, with CatBoost set to 1,000 iterations, a depth of 6, and a learning rate of 0.1; Random Forest with 100 trees; and LSTM with 64 hidden units and 2 layers.

**CEEMDAN-XGBoost** (Lu et al., 2020b): This model integrates CEEMDAN for data decomposition with XGBoost for prediction. The implementation uses the XGBoost library and the CEEMDAN class from PyEMD, employing a squared error loss function with 100 estimators.

**LSTM** (Graves, 2012): A specialized form of RNN featuring input, output, and forget gates. Implemented using PyTorch, with a hidden layer dimension of 64 and two layers.

**GRU** (Cho et al., 2014): A simplified variant of LSTM, omitting the forget gates and using update and reset gates. Implemented in PyTorch with a hidden layer dimension of 64 and two layers.

**FDL** (Gu et al., 2019): Combines entropy-based gray correlation analysis with LSTM and GRU for lane-level prediction. Implemented in PyTorch, with both LSTM and GRU having a hidden dimension of 64 and two layers.

**TM-CNN** (Ke et al., 2020): This model transforms traffic speed and volume data into matrices for prediction. It is implemented in PyTorch as a single-stream (using only speed or volume), multi-channel convolutional network to ensure fairness.

**MDL** (Lu et al., 2020a): Combines ConvLSTM, convolutional layers, and dense layers for lane-based dynamic traffic prediction. The model is implemented using the authors' code[8], with adjustments made for fairness, similar to TM-CNN.

**CNN-LSTM** (Ma et al., 2020): Enhances short-term traffic prediction by integrating CNN for lane analysis. Implemented in PyTorch with a hidden layer dimension of 16 and one LSTM layer.

**HGCN** (Zhou et al., 2022) and **DGCN** (Wang et al., 2021): Both models, developed by the same author, utilize identical formulas and methodologies, integrating spatial dependency analysis, data fusion, and temporal attention. Implemented in

---

[5]https://github.com/HKUDS/FlashST

[6]https://github.com/HKUDS/GPT-ST

[7]https://github.com/ShuhaoLii/McgVAE

[8]https://github.com/lwqs93/MDL

PyTorch, with heterogeneous data excluded to ensure fairness.

**GCN-GRU** (Li et al., 2023a): Combines a GCN with a data-driven adjacency matrix and GRU. Implemented in PyTorch, with the GCN output dimension set to 16, and the GRU configured with 64 hidden units and two layers.

**ST-AFN** (Shen et al., 2021): Features a speed process network, spatial encoder, and temporal decoder with an embedded attention mechanism. Implemented using the authors' code[9].

**STA-ED** (Zheng et al., 2022): Utilizes an encoder-decoder architecture with LSTM and a two-stage attention mechanism. Implemented in PyTorch, with a 64-unit hidden layer.

**STMGG** (Wang et al., 2021): Leverages visibility graphs, spatial topological graphs, an attention-based gated mechanism, and Seq2Seq for lane-level traffic prediction. Implemented in PyTorch with a 64-unit hidden layer.

**ST-ABC** (Li et al., 2024a): ST-ABC is a spatio-temporal convolutional network that leverages an improved attention mechanism and fully convolutional architecture to efficiently model spatio-temporal correlations in complex dynamic scenarios. The model enhances prediction performance by utilizing a finely designed attention mechanism for spatio-temporal data. The implementation is based on the official code provided by the authors[10].

### J.4. Road-Level Task Models

**DCRNN** (Li et al., 2018): Simulates diffusion processes on traffic graphs, combining spatio-temporal dynamics, bidirectional walks, and an encoder-decoder architecture. Implemented using the authors' code[11].

**STGCN** (Yu et al., 2018): Efficiently models traffic networks on graphs with a fully convolutional structure, enabling faster training and reducing parameter count. Implemented using the authors' code[12].

**MTGNN** (Wu et al., 2020): Introduces a graph neural network framework for multivariate time series, automatically extracting variable relations and capturing spatio-temporal dependencies through innovative layers. Implemented using the authors' code[13].

**ASTGCN** (Guo et al., 2019): An attention-based spatio-temporal graph convolutional network (ASTGCN) that models recent, daily, and weekly traffic dependencies using a space-time attention mechanism and graph convolution. Implemented using the authors' code[14].

**GraphWaveNet** (Wu et al., 2019): Utilizes an adaptive dependency matrix and node embedding to capture spatial data dependencies, processing long sequences with dilated one-dimensional convolutions. Implemented using the authors' code[15].

**STSGCN** (Song et al., 2020): Captures localized spatio-temporal correlations using a synchronous modeling mechanism with modules for different time periods. Implemented using the authors' code[16].

**AGCRN** (Bai et al., 2020): Combines adaptive learning modules with recurrent networks to autonomously capture detailed spatio-temporal traffic correlations. Implemented using the authors' code[17].

**STGODE** (Fang et al., 2021): Models spatio-temporal dynamics with tensor-based ordinary differential equations (ODEs), constructing deeper networks that leverage these features. Implemented using the authors' code[18].

**STAEformer** (Liu et al., 2023a): Builds a spatio-temporal autoencoder with transformers, utilizing temporal-aware attention and spatial-enhanced blocks to capture complex dependencies. Achieves strong traffic prediction performance with fewer

---

[9]https://github.com/MCyutou/ST-AFN
[10]https://github.com/ShuhaoLii/ICDE2024
[11]https://github.com/liyaguang/DCRNN
[12]https://github.com/VeritasYin/STGCN_IJCAI-18
[13]https://github.com/nnzhan/MTGNN
[14]https://github.com/wanhuaiyu/ASTGCN
[15]https://github.com/nnzhan/Graph-WaveNet
[16]https://github.com/Davidham3/STSGCN
[17]https://github.com/LeiBAI/AGCRN
[18]https://github.com/square-coder/STGODE

parameters. Implemented using the authors' code[19].

**MegaCRN** (Jiang et al., 2023): Integrates a meta graph learner with a GCRN encoder-decoder architecture, effectively handling varied road patterns and adapting to abnormal traffic conditions. Implemented using the authors' code[20].

**Time-Mixer** (Wang et al., 2024): An MLP-based model that uses multiscale-mixing to separate and combine seasonal and trend components across different scales, improving time series forecasting. It employs PDM for past data decomposition and FMM for enhancing future predictions. Implemented using the authors' code[21].

**iTransformer** (Liu et al., 2023c): Reframes time series forecasting as a sequence matching problem and proposes a concise linear transformer with downsampling. Achieves high efficiency and accuracy without traditional positional encoding. Implemented using the authors' code[22].

---

[19] https://github.com/XDZhelheim/STAEformer
[20] https://github.com/deepkashiwa20/MegaCRN
[21] https://github.com/kwuking/TimeMixer
[22] https://github.com/thuml/iTransformer

