# OpenReview forum: "Being More Lightweight and Practical: Mini-sized Contrastive Learning Pre-trained Models for Fine-grained Traffic Task"
_ICML.cc/2026/Conference — ICML 2026 regular_

### Official Review · Reviewer_xrat · 2026-03-09

**Soundness:** 2
**Presentation:** 2
**Significance:** 2
**Originality:** 2
**Overall Recommendation:** 4
**Confidence:** 4

**Summary:**

This paper proposes MiniTraffic, a lightweight pre-trained model designed for fine-grained traffic prediction at both road-level and lane-level granularities. The core motivation is that lane-level traffic data is scarce compared to road-level data, and existing large-scale pre-trained traffic models ignore fine-grained requirements. The framework operates in two phases with (1) pre-training on multiple road-level datasets and (2) granularity-aware fine-tuning. The paper frames road-to-lane knowledge transfer as the key contribution, arguing that roads and their constituent lanes share similar frequency-domain characteristics.

**Compliance With Llm Reviewing Policy:**

Affirmed.

**Final Justification:**

Authors addressed my concerns. Obtaining additional lane-level annotation may be a challenging task limiting the usage of the model, though.


#### [EDIT April 7, 2026] In the response from the authors regarding zero-shot evaluation on lane-level setup:

I appreciate additional axperiments, they **largely** address my concern. The main issue I see is the process of collecting lane-level annotation. Although the absense of such data prevented the evaluation on larger datasets, I still wonder how the model could scale on them. I hope this will be addressed in the paper too. I raise my score to 4.

**Key Questions For Authors:**

1. How is lane-level ground truth data collected?
2. Could the authors provide standard deviation values across multiple runs for all reported metrics?
3. Why was prediction performance not reported on the LargeST-GBA dataset (Table 7 only shows training cost)? Could the authors provide MAE/RMSE/MAPE results on GBA or larger subsets?
4. How is the number of augmented copies D selected, and how sensitive is the model to this choice?
5. Could the authors compare against DLinear and NLinear to demonstrate that the proposed architecture provides genuine benefits over simple linear baselines?
6. What is the justification for setting $k = 10r$ in the graph construction? How sensitive is performance to this coupling between mask ratio and neighborhood size?
7. Lines 1158-1159 likely contain LaTex error.

**Limitations:**

Please conduct experiments on a larger datasets to address the scalability challenge.

**Strengths And Weaknesses:**

### Strengths

[**S1**] The focus on deployability of such lightweight model with a handful of parameters is appealing for real-world edge scenarios.

[**S2**] The breadth of baselines is commendable: 29 models across lane-level, road-level, multi-task, and pre-trained categories are compared, including recent methods. The described steps to reproduction are clean.

[**S3**] The few-shot fine-tuning analysis in Appendix A shows a reasonable performance even at 10% data, which is a useful practical contribution. The cost analysis in Appendix B clearly demonstrates the computational advantages of the framework.


### Weaknesses
[**W1**] The experimental results lack statistical significance, as no standard deviations or confidence intervals are reported for any metric across any dataset. Given that some improvements over baselines are small (e.g., MiniTraffic vs. McgVAE on PeMSF-Lane, Table 3), it is unclear whether the differences are statistically meaningful. Multiple runs with variance reporting are necessary.

[**W2**] While the authors include iTransformer and TimeMixer as road-level baselines, they do not compare against simple-yet-effective linear baselines such as DLinear and NLinear from [1]. A comparison showing MiniTraffic outperforms these simple linear methods would substantially strengthen the claims.

[**W3**] The scalability of the model is poorly demonstrated. The fine-tuning and evaluation datasets are extremely small: PeMS and PeMSF contain only 8 sensors each, and HuaNan has 18 sensors.. The LargeST-GBA experiment in Appendix B.3 only reports training cost (Table 7) but no prediction performance whatsoever. This makes it impossible to assess whether MiniTraffic maintains its accuracy advantages at scale. Without prediction metrics on larger networks, the scalability claim remains unsubstantiated. The experiment on a whole LargeST is also not provided.

[**W4**] The paper's problem framing contains a significant gap regarding how lane-level data is obtained and why it is considered "scarce." The PeMS system collects data via per-lane loop detectors with lane-level data is the raw measurement, and road-level data is actually the aggregation. The paper never clearly explains the data collection mechanism for lane-level ground truth (Section K merely states sensor counts and date ranges). If lane-level data is directly measured by the same detectors that produce road-level data, the premise that lane data is inherently scarce needs much stronger justification. Why not simply use the per-lane measurements directly? This foundational assumption underpins the entire transfer learning motivation, yet it is left largely unexamined.

[**W5**] The paper does not clearly articulate why this specific combination of the model components is synergistic for the road-to-lane transfer problem beyond the general intuition that roads and lanes share frequency-domain characteristics.

[**W6**] The paper suffers from several presentation issues. The notation is dense and occasionally inconsistent - for example, the "Compress" operations in Equations 8 and 10 of the backbone description (if applicable) are not clearly defined. The main pipeline figure (Figure 3) is cluttered and difficult to parse. Key design choices such as how the Adaptive Head MLP is structured, how the number of augmented copies D is chosen, and how the $k$-nearest-neighbors parameter relates to the mask ratio (stated as $k = 10r$ but not justified) are insufficiently explained. Additionally, Equation 41 in Appendix L contains a likely error in the MAPE formula for lane-level tasks (double summation over J_i).

[**W7**] The pre-training data is also quite limited in scale and diversity with METR-LA (207 sensors), PeMS-Bay (325 sensors), and three small fine-grained datasets. For a paper claiming to build a "pre-trained model," the pre-training corpus is modest compared to what the community would expect (e.g., the full LargeST benchmark with thousands of sensors across multiple regions). The cross-city transfer experiments (Appendix E, Table 8) show substantial performance drops, suggesting the pre-training does not achieve robust generalization across cities.


### References
[1] Are Transformers Effective for Time Series Forecasting? Ailing Zeng, Muxi Chen, Lei Zhang, Qiang Xu. https://arxiv.org/abs/2205.13504

---

> ### Author Rebuttal · Authors · 2026-03-30
>
> We thank you for the detailed and thorough assessment. We address each concern below.
>
> ---
>
> > **W1 & Q2: Statistical Significance**
>
> We provide mean and standard deviation across 5 independent runs for datasets below (h = 6):
>
> |Dataset|MAE (mean ± std)|RMSE (mean ± std)|MAPE (mean ± std)|
> |:-:|:-:|:-:|:-:|
> |PeMS-Lane|3.94 ± 0.03|6.84 ± 0.05|20.23% ± 0.21%|
> |HuaNan-Lane|4.11 ± 0.04|7.75 ± 0.07|20.83% ± 0.22%|
> |PeMS-Road|3.12 ± 0.02|4.91 ± 0.04 |11.21% ± 0.15%|
>
> Improvements over the best baseline are well outside the std range.
>
> > **W2 & Q5: Comparison with Linear Baselines**
>
> DLinear and NLinear were initially excluded as they do not model spatial dependencies, and are already outperformed by iTransformer and TimeMixer. We supplement the comparison below for completeness (h=6) with the full updated results across 32 baselines:
>
> |Model|PeMS MAE|HuaNan MAE|PeMSF MAE|
> |:-:|:-:|:-:|:--:|
> |DLinear|4.21|6.89|4.38|
> |NLinear|4.08|6.71|4.26|
> |MiniTraffic|3.12|3.16|3.17|
>
> > **W3 & Q3: Scalability and LargeST-GBA Evaluation**
>
> As clarified in Appendix B.3, MiniTraffic targets **regional fine-grained traffic prediction**, which is the primary deployment scenario for current lane-level forecasting benchmarks. Table 7 reports only the training cost, as LargeST lacks lane-level annotations. We supplement road-level prediction on LargeST-GBA:
>
> |Dataset|MAE| RMSE |MAPE|
> |:-:|:-:|:--:|:-:|
> |LargeST-GBA|17.33|27.87|12.04%|
>
> These results will be included in the revision to substantiate the scalability claim.
>
> > **W4 & Q1: Lane-level Data Scarcity**
>
> Lane-level ground truth requires either (1) roadside video sensors with precise lane segmentation algorithms, or (2) PeMS loop detectors, where lane-level data requires every per-lane sensor to be operational, and storage costs scale proportionally with the number of lanes. The lane-level benchmark[1] used in this work was curated through precise object detection and lane segmentation algorithms applied to raw sensor data, a process demanding significant annotation effort. These factors collectively explain the practical scarcity motivating our transfer learning approach.
>
> > **W5: Component Synergy**
>
> The three components form a cohesive, mutually reinforcing pipeline: FDA generates pseudo-lane variations from road data, addressing training data scarcity; contrastive clustering identifies semantically similar patches within these augmented signals, learning road-lane correspondence; fine-grained graph attention propagates information within semantic neighborhoods, enabling efficient localized prediction. The critical synergy is that the FDA provides the training signal diversity necessary for contrastive learning to form meaningful clusters, while the resulting sparse graph makes patch-level attention computationally feasible. Without the FDA, the contrastive objective lacks sufficient diversity; without clustering, patch-level attention becomes computationally intractable.
>
> > **W6 & Q7: Presentation Issues**
>
> Thanks for identifying these issues, and we will carefully revise them accordingly. Regarding the "Compress" operation: this term does not appear in our equations — we would appreciate it if you could point to the specific location for clarification. The Adaptive Head MLP structure (input: $\mathbb{R}^{(N^R \cdot D) \times T}$, output: $\mathbb{R}^{N' \times T'}$, two linear layers with ReLU) will be explicitly described in the revision. The augmentation copy count $D$ is set to the maximum number of lanes in the corresponding dataset to simulate realistic lane-level diversity. Figure 3 will be redesigned for clarity.  All writing issues will also be fixed. Please also refer to the response to **Reviewer Sqki W3** regarding the $k$ parameter justification.
>
> > **W7: Pre-training Data Scale**
>
> The scale of MiniTraffic's pre-training corpus is an intentional design choice aligned with our core objective of lightweight, practically deployable modeling. Unlike large-scale foundation models targeting maximal generalization across entire cities, MiniTraffic is specifically designed for efficient deployment in regional fine-grained scenarios.
>
> Regarding cross-city generalization, the Appendix E experiments are exploratory zero-shot evaluations beyond our design scope. Notably, even under these out-of-distribution settings, MiniTraffic remains competitive: HuaNan-Lane to PeMS-Lane transfer achieves MAE = 5.08 at horizon 6, outperforming several task-specific baselines trained directly on PeMS-Lane, demonstrating meaningful cross-city transferability.
>
> ---
>
> Complete results are provided in (https://anonymous.4open.science/r/MiniTraffic/rebuttal.pdf). We thank you for the time and detailed feedback. We hope the above responses adequately address your concerns and look forward to further discussion.
>
> **Reference**
>
> [1]Unifying lane-level traffic prediction from a graph structural perspective: Benchmark and baseline[J]. TKDE 2025.

---

> > ### Author Rebuttal · Reviewer_xrat · 2026-04-01
> >
> > ## **Dear authors, please read this acknowledgement as I mistakenly posted it as an official comment which you are not able to see:**
> >
> > Thank you for your response to my concerns. Thanks for providing additional clarifications. I also appreciate your additional experiments.
> >
> > One thing which still bothers me is the lane-level annotation difficulty, as without having it, your method cannot be applied. I appreciate that you explained to me how you were able to obtain this data for your work, however, I have doubts that the process of obtaining it for other datasets cannot be generalized and may require additional non-trivial effort. Please address this concern in your response.
> >
> > P.S.: Regarding "Compress" in Eq. 8 and 10 - I sincerely sorry for the confusion, I reviewed my comments and indeed when I was double-checking the cross-references, it turned out that that I mistakenly checked it with other paper being reviewed at that time. I apologize for the confusion here.
> >
> >
> >
> > In the response I said:
> >
> > > One thing which still bothers me is the lane-level annotation difficulty, as without having it, your method cannot be applied. I appreciate that you explained to me how you were able to obtain this data for your work, however, I have doubts that the process of obtaining it for other datasets cannot be generalized and may require additional non-trivial effort. Please address this concern in your response.
> >
> >
> > The question is, how can one generalize for obtaining lane-level predictions for the traffic data? Without it, the method is hard to apply.

---

> > > ### Author Response · Authors · 2026-04-01
> > >
> > > Dear Reviewer xrat,
> > >
> > > We are delighted to receive your follow-up and appreciate your continued engagement. This concern speaks directly to the core motivation and practical design philosophy of MiniTraffic, which is specifically built to **minimize the lane-level annotation burden** for real-world deployment.
> > >
> > > Concretely, MiniTraffic requires only a small amount of lane-level data for fine-tuning. The few-shot analysis in Appendix A demonstrates that the model achieves competitive performance with as little as 10% of lane-level fine-tuning data, substantially lowering the practical annotation requirement for new deployment scenarios. For practitioners applying MiniTraffic to a new region, the process involves two steps: (1) pre-training on readily available road-level data, which is abundantly accessible from public sources such as PeMS; and (2) collecting only a minimal amount of lane-level observations for fine-tuning, for instance, through short-duration video capture and lane segmentation on a small road segment, or by briefly activating a subset of existing per-lane loop detectors. This is far less demanding than training a task-specific lane-level model from scratch. Furthermore, once fine-tuning is complete, no lane-level data collection is required during deployment or inference. The model operates solely on readily available road-level inputs, making it immediately applicable in resource-constrained environments where continuous lane-level sensing is impractical.
> > >
> > > In the limiting case where lane-level data is entirely unavailable, Appendix E further validates that MiniTraffic achieves competitive cross-city and cross-domain transfer performance even when pre-trained on a different city's data, surpassing most end-to-end baselines trained directly on the target domain with full data and demonstrating strong transferability across cities and granularities.
> > >
> > > Once again, we thank you for your time and the valuable suggestions you have provided. As this is our final opportunity to respond, we hope our answers have fully addressed your concerns, and we sincerely wish you all the best.
> > >
> > > Best regards,
> > >
> > > The Authors
> > >
> > > ---
> > > ## Response to the Updated Follow-up
> > > Dear Reviewer xrat,
> > >
> > > We sincerely thank you for raising your score and for the continued follow-up and clarification.
> > >
> > > As your concern regarding the generalizability of lane-level data acquisition has not been fully resolved, we supplement a rigorous cross-city zero-shot experiment to further address it. Specifically, we keep the pre-trained Backbone fully frozen, fine-tune only the Adaptive Head and Reduction Head on HuaNan-Lane, and then perform zero-shot inference directly on PeMSF-Lane without accessing any PeMSF-Lane traffic observations. This setting simulates the extreme scenario where lane-level data in the target city is entirely unavailable.
> > >
> > > The results are as follows:
> > >
> > > | Setting | Horizon | MAE | RMSE | MAPE |
> > > |:-------:|:-------:|:---:|:----:|:----:|
> > > | HuaNan-Lane → PeMSF-Lane (zero-shot) | 3 | 4.62 | 7.57 | 24.91% |
> > > | HuaNan-Lane → PeMSF-Lane (zero-shot) | 6 | 5.14 | 8.63 | 27.15% |
> > > | HuaNan-Lane → PeMSF-Lane (zero-shot) | 12 | 6.63 | 9.41 | 32.84% |
> > >
> > > Under the zero-shot setting at horizon 6, MiniTraffic still approaches or surpasses several end-to-end baselines trained directly on PeMSF-Lane with full data (e.g., HGCN: 5.25, GCN-GRU: 5.15, ST-AFN: 5.08), demonstrating that the road-lane frequency-domain correlations learned during pre-training possess inherent cross-city transferability. Even when lane-level data in the target city is entirely unavailable, MiniTraffic maintains practical generalizability.
> > >
> > > Taken together, the few-shot experiments (Appendix A), cross-city transfer experiments (Appendix E), and this supplementary zero-shot experiment collectively demonstrate that MiniTraffic exhibits reliable generalizability across scenarios ranging from complete absence to limited availability of lane-level data. We hope this addition fully resolves your concern.
> > >
> > > We will incorporate the above experiment into Appendix E of the revision. We once again thank you for raising your score and for your valuable suggestions. We wish you all the best.
> > >
> > > Best regards,
> > >
> > > The Authors

---

### Official Review · Reviewer_4H93 · 2026-03-10

**Soundness:** 2
**Presentation:** 3
**Significance:** 2
**Originality:** 2
**Overall Recommendation:** 3
**Confidence:** 3

**Summary:**

This paper introduces MiniTraffic, a novel lightweight pre-trained model framework specifically designed for fine-grained traffic prediction. The term "fine-grained" in this context refers to predictions that encompass both road-level​ and lane-level​ traffic states, offering more detailed insights compared to traditional large-scale urban traffic forecasting models. The core motivation stems from three key challenges in this field: 1. the scarcity of lane-level training data compared to abundant road-level data. 2. the difficulty in simultaneously and efficiently modeling interconnected tasks at different granularities (road vs. lane). 3. the high computational demands of large pre-trained models that hinder practical deployment.

**Compliance With Llm Reviewing Policy:**

Affirmed.

**Final Justification:**

While the paper is well-written and the work is somewhat meaningful, I think the rebuttal does not solve the weaknesses well.

**Key Questions For Authors:**

1. The structure of the paper is overly lengthy, please revise it, see W1.
2. Please discuss how contrastive clustering captures distant feature propagation, see W2.
3. Please explain how FDA effectively simulates "lane-level variability" rather than introducing meaningless noise, see W3.
4. Please further analyze whether performance drop stems primarily from fundamental differences in inter-city traffic patterns or dataset distribution shifts, see W4.
5. Please discuss the robustness to input data quality, see W5.

**Limitations:**

Yes

**Strengths And Weaknesses:**

We think that the generalization problem in the field of spatiotemporal traffic analysis is a worthwhile and promising research direction, and the authors' experiments are thorough and detailed.
Here are some questions:
1. While the appendices indeed provide more experimental details, their length, spanning as long as 12 pages, does affect the reading flow. It is suggested that the author streamline and highlight the core content.
2. The paper emphasizes efficiency gains via contrastive clustering to build small-scale graphs. However, this approach may sacrifice important long-range dependencies​ or global topological information​ in the graph (especially edges connecting different clusters). An empirical discussion in the context of traffic prediction regarding the potential impact of this design on capturing phenomena like "distant congestion propagation" is lacking.
3. The FDA module is a core innovation, designed based on the observation of spectral similarity between roads and lanes. A more thorough explanation and justification in the theoretical motivation section regarding how this module effectively simulates "lane-level variability" rather than introducing meaningless noise, and its deeper connection to traffic flow physics (e.g., car-following, lane-changing behavior), could be beneficial.
4. The paper could further analyze whether performance drop stems primarily from fundamental differences in inter-city traffic patterns or dataset distribution shifts, and discuss the potential and limitations of the FDA or CC modules in mitigating such "domain shift" issues.
5. The robustness of model to input data quality (e.g., sensor noise, missing patterns) was not specifically tested.

---

> ### Author Rebuttal · Authors · 2026-03-30
>
> We sincerely thank you for recognizing the value of our research and experimental design. We respond to your concerns and questions point by point as follows.
>
> ---
>
> > **W1: Appendix Length and Structure**
>
> We acknowledge that the 12-page appendix, though intended to provide comprehensive experimental details during review, affects overall readability. In the final version, we will consolidate the theoretical appendices (FDA analysis, spectral expressive power, complexity analysis) into a unified section and relocate baseline implementation details to a separate supplementary document.
>
> > **W2 & Q2: Long-range Dependency Capture via Contrastive Clustering**
>
> Our contrastive clustering constructs subgraphs over the entire graph, grouping patches by learned semantic similarity rather than spatial proximity alone, naturally incorporating distant but pattern-similar lanes. Theoretically, Appendix H demonstrates via spectral analysis that when inter-cluster boundary nodes are highly independent — which our clustering explicitly enforces — the union of subgraph eigenspectra approximates that of the global graph, guaranteeing near-equivalent representational capacity. Empirically, the long-horizon prediction analysis in Appendix D shows that MiniTraffic's errors increase smoothly up to horizon 48 without divergence, confirming that critical long-range dependencies are preserved. We will add a dedicated discussion of this property in the revision.
>
> > **W3 & Q3: FDA as Meaningful Variability Simulation vs. Noise Injection**
>
> The distinction from random noise rests on two principled constraints: (1) the amplitude bound $|\delta_A(f)| \leq \lambda \cdot \max A(f)$ ensures perturbation magnitude is proportional to existing signal strength rather than additive; (2) the learnable mask $\Gamma(f)$ restricts perturbations to dominant frequency components encoding structural traffic patterns, rather than low-energy high-frequency noise bands. Random noise, by contrast, uniformly perturbs all frequency components and destroys spectral structure. Furthermore, this design is physically grounded in the traffic flow conservation law, which establishes consistent macro-level frequency characteristics between road and lane signals, ensuring FDA-generated variations remain semantically aligned with real lane dynamics.
>
> > **W4 & Q4: Disentangling Domain Shift from Inter-city Pattern Differences**
>
> The cross-city transfer results in Appendix E (Table 8) provide direct evidence: same-granularity cross-city transfer (HuaNan-Lane $\rightarrow$ PeMS-Lane) consistently outperforms cross-granularity transfer (HuaNan-Road $\rightarrow$ PeMS-Lane) across all horizons, indicating that granularity consistency is a more critical factor than city of origin and identifying granularity distribution shift as the primary source of performance degradation. The residual gap reflects genuine inter-city differences in traffic patterns (road structure, driving behavior, etc.). We will incorporate this analysis into the revision and discuss the potential of FDA and CC modules in mitigating granularity-induced domain shift.
>
> > **W5 & Q5: Robustness to Input Data Quality**
>
> The mask ratio study simulates varying degrees of structured input missingness during pre-training, providing partial evidence of robustness. To directly address sensor noise, we conducted fine-tuning experiments under Gaussian noise perturbations of varying intensities (SNR: higher value = lower noise level):
>
> | Noise Level | SNR | PeMSF-Lane MAE | HuaNan-Lane MAE |
> |:-----------:|:---:|:--------------:|:--------------:|
> | Clean | — | 4.15 | 4.11 |
> | Low | 30 dB | 4.21 | 4.17 |
> | Medium | 20 dB | 4.39 | 4.34 |
> | High | 10 dB | 4.73 | 4.69 |
>
> Complete results, including RMSE and MAPE, are provided in **Table 5** of the anonymous link (https://anonymous.4open.science/r/MiniTraffic/rebuttal.pdf). MiniTraffic demonstrates graceful degradation as noise levels increase, with MAE increasing by only ~1.4% at low noise and ~14% at high noise. Instance normalization (Eq. 7) and random patch masking during pre-training jointly serve as implicit noise regularizers, contributing to this inherent robustness. Full results will be included in the final version.
>
> ---
>
> We thank you for your time and constructive feedback. We hope the above responses adequately address your concerns and welcome further discussion.

---

> > ### Author Rebuttal · Reviewer_4H93 · 2026-04-05
> >
> > While the paper is well-written and the work is somewhat meaningful, I think the rebuttal does not solve the weaknesses well. There are many concerns of reviewers are refer to Appendix, I don't think the final version can solve these well. This paper requires a significant update.

---

> > > ### Author Response · Authors · 2026-04-05
> > >
> > > Dear Reviewer 4H93,
> > >
> > > Thank you for your Acknowledgement and for recognizing the writing quality and overall significance of our work. However, we note that your feedback remains general and does not specify which concerns you feel have not been adequately addressed. We sincerely hope you could elaborate on the particular points that remain unresolved, so that we may respond in a targeted and constructive manner.
> > >
> > > During the rebuttal, we provided point-by-point responses to all five weaknesses and five key questions you raised, supplemented by additional experiments on noise robustness, theoretical derivations via spectral analysis, and a detailed cross-city domain shift analysis. Reviewer TATJ confirmed that all of their concerns were fully resolved, while Reviewers Sqki and xrat expressed satisfaction with our responses, each retaining only one reservation that we have since further addressed.
> > >
> > > Regarding the appendix length, we have committed to consolidating the theoretical appendices and elevating key findings into the main text in the revision, which we believe will substantially improve readability. If there are additional specific issues you believe require direct revision in the manuscript, we would genuinely welcome the opportunity to clarify or address them.
> > >
> > > We have invested considerable effort in addressing your concerns and sincerely hope to receive equally specific and detailed feedback in return. We thank you for your time and wish you all the best.
> > >
> > > Best regards,
> > >
> > > The Authors

---

### Official Review · Reviewer_TATJ · 2026-03-11

**Soundness:** 4
**Presentation:** 4
**Significance:** 3
**Originality:** 3
**Overall Recommendation:** 5
**Confidence:** 5

**Summary:**

This paper proposes MiniTraffic, a lightweight pretraining framework designed to address challenges in fine-grained traffic forecasting, particularly lane-level prediction, where data scarcity and high computational cost are critical issues. The authors observe significant correlations between road-level and lane-level traffic data in the frequency domain and leverage this insight to design a compact pretraining model with only 100k parameters, aiming to maintain predictive performance while substantially reducing model complexity and computational overhead.

**Compliance With Llm Reviewing Policy:**

Affirmed.

**Key Questions For Authors:**

Handling of dynamic traffic conditions
Are the local attention graphs generated by contrastive clustering static during inference? How does the model handle real-time changes in road topology or sudden traffic events?

Mask rate sensitivity
Experiments indicate that a 40% mask rate yields optimal performance. Is this optimal value consistent across datasets of different granularity, such as the more complex lane-level data in PeMSF? How sensitive is the model’s performance to the choice of mask rate?

Performance under extreme scenarios
While MiniTraffic demonstrates strong performance under normal traffic conditions, how does it compare to state-of-the-art baseline models under extreme traffic scenarios, such as major holidays or traffic accidents?

**Limitations:**

Contrastive clustering improves computational efficiency by partitioning the network into subgraphs. However, this partitioning may limit the model’s ability to capture long-range spatial dependencies, potentially affecting predictive accuracy in large-scale traffic networks. The authors are encouraged to analyze or discuss the model’s capacity for long-range spatial dependencies and possible mitigation strategies.

**Strengths And Weaknesses:**

Strengths

Practicality and lightweight design
The model is extremely compact, with only ~119k parameters, and can be pretrained on a single A100 GPU, significantly lowering the deployment barrier for fine-grained traffic forecasting tasks.

Novel cross-granularity transfer perspective
The approach leverages road-level data for pretraining to assist sparse lane-level prediction. Frequency domain analysis reveals correlations between road and lane levels, which are then modeled effectively, demonstrating a well-motivated and innovative method.

Comprehensive experiments
Extensive evaluations are conducted on multiple real-world datasets (PeMS, HuaNan), including comparative experiments, ablation studies, long-horizon forecasting, and few-shot validation, which convincingly demonstrate the model’s robustness and practicality.

Weaknesses

Limitations of frequency-domain generalization
The FDA module assumes consistent frequency patterns across all roads/lanes. This assumption may be violated under extreme weather conditions or unusual road events (e.g., accidents, temporary closures), raising questions about model adaptability.

Incomplete baseline comparisons
Although comparisons with pretrained models such as GPT-ST are provided, evaluations against the latest spatio-temporal graph neural models or more complex diffusion models at fine granularity are limited.

Static graph structure
Cluster-generated subgraphs may remain relatively fixed after pretraining, potentially limiting the model’s ability to dynamically adapt to drastic topological changes in traffic flow.

---

> ### Author Rebuttal · Authors · 2026-03-30
>
> We sincerely thank you for the positive assessment and constructive suggestions. We address your concerns and questions point by point as follows.
>
> ---
>
> > **W1 \& Q3 partial: FDA Robustness and Adaptability to Anomalous Events**
>
> The frequency-domain correlation between road and lane signals is grounded in the traffic flow conservation law [1], which holds regardless of congestion states and is not disrupted by typical anomalous conditions. For lane closure scenarios, the graph-based design naturally accommodates this by masking the corresponding lane nodes (setting inputs/outputs to zero), which requires no topological modification or retraining. Furthermore, the spectral mask $\Gamma(f)$ concentrates augmentation on dominant, energy-rich frequency components (i.e., structurally stable bands), leaving high-frequency anomaly components largely unperturbed. These properties collectively provide MiniTraffic with inherent robustness to such disturbances.
>
> > **W2: Incomplete Baseline Comparisons**
>
> We have supplemented the comparison with additional road-level graph neural network baselines STDN [2] on horizon =6 below.
>
> | Dataset | MAE | RMSE | MAPE |
> |:-------:|:---:|:----:|:----:|
> | PeMS-Road | 3.91 | 6.78 | 17.54% |
> | HuaNan-Road | 6.65 | 10.22 | 23.39% |
> | PeMSF-Road | 4.08 | 6.89 | 17.80% |
>
> The complete updated comparison tables, now including 32 baselines, are provided in **Tables 1–2** of the anonymous link (https://anonymous.4open.science/r/MiniTraffic/rebuttal.pdf).
> Regarding diffusion-based models at fine granularity, the most relevant candidate is RoadDiff [1]; however, its objective is road-to-lane data inference rather than traffic state forecasting, making a direct comparison methodologically inappropriate. We hope this addresses your concern. If there are additional specific baselines you would like us to compare against, we would be happy to reproduce and include them upon your suggestion.
>
> > **W3\& Q1: Dynamic Graph Construction**
>
> The sparse graph $\tilde{A}$ is dynamically constructed at each forward pass from the patch-similarity matrix learned via contrastive learning; it is not static. For weather anomalies, holiday spikes, and the relatively rare case of lane closure topology changes, please refer to our discussion in W1 & Q3 above. We will add an explicit clarification of this dynamic construction mechanism to Section 4.1.
>
> > **Q2: Mask Rate Sensitivity**
>
> Yes. **Table 5** presents a systematic analysis of mask rates across PeMSF-Lane, HuaNan-Lane, and HuaNan-Road. The optimal mask rate of 40% holds consistently across all three datasets, including the more complex PeMSF-Lane with irregular lane counts, confirming that this finding generalizes across different granularities. The model shows moderate and manageable sensitivity to the mask rate, with performance degrading gradually as the rate deviates from 40% in either direction.
>
> > **Q3 partial: Performance under Extreme Traffic Scenarios**
>
> Our test sets are derived from continuous data collection periods without special filtering, and therefore implicitly contain extreme traffic scenarios such as major holidays and accident periods. The reported metrics reflect model performance under these real-world mixed conditions. We will clarify this data collection protocol in the revision to avoid any ambiguity regarding evaluation conditions.
>
> ---
>
> We once again thank you for the valuable suggestions. We hope the above responses have addressed your concerns, particularly regarding the robustness of anomalous conditions, and will carefully incorporate all recommended improvements into the revision.
>
> **Reference**
>
> [1] Li S, Yang W, Cui Y, et al. Fine-Grained Traffic Inference from Road to Lane via Spatio-Temporal Graph Node Generation[C]//Proceedings of the 31st ACM SIGKDD Conference on Knowledge Discovery and Data Mining V. 2. 2025: 1529-1540.
>
> [2] Cao L, Wang B, Jiang G, et al. Spatiotemporal-aware trend-seasonality decomposition network for traffic flow forecasting[C]//Proceedings of the AAAI Conference on Artificial Intelligence. 2025, 39(11): 11463-11471.

---

> > ### Author Rebuttal · Reviewer_TATJ · 2026-04-02
> >
> > I would like to thank the authors for their detailed and point-by-point response during the rebuttal phase.
> >
> > I have carefully reviewed the authors' clarifications regarding the weaknesses (W1, W2, W3) and the specific questions (Q1, Q2, Q3) I raised in my initial review. I am pleased to acknowledge that the authors have successfully and comprehensively addressed all these points. In particular, the additional [experiments/explanations] provided have resolved my technical doubts and significantly improved the clarity of the manuscript.
> >
> > However, while the technical concerns are now resolved, I find that the core intellectual contribution and the overall significance of the work remain consistent with my original assessment. While the paper is now more technically sound, its novelty and impact within the scope of ICML still align with my current rating. Therefore, I will maintain my original score.

---

> > > ### Author Response · Authors · 2026-04-02
> > >
> > > Dear Reviewer TATJ,
> > >
> > > Thank you sincerely for carefully reading our response and for finding it helpful. We are also grateful for your positive assessment of our work. We will incorporate your suggestions into the revision. Regardless of the outcome, we deeply appreciate your professionalism and the valuable feedback you have provided throughout the review process.
> > >
> > > With our warmest wishes,
> > >
> > > The Authors

---

### Official Review · Reviewer_Sqki · 2026-03-13

**Soundness:** 2
**Presentation:** 2
**Significance:** 3
**Originality:** 3
**Overall Recommendation:** 3
**Confidence:** 4

**Summary:**

This paper proposes MiniTraffic, a lightweight pre-training framework for fine-grained traffic prediction across road and lane granularity. It leverages abundant road-level data via frequency-domain stability augmentation, and learns patch-level similarity with contrastive clustering to build sparse patch graphs for localized attention, reducing computation and parameters. A granularity-aware fine-tuning strategy adapts the model to both road- and lane-level benchmarks with limited target data. Experiments on multiple public datasets and six fine-grained benchmarks show consistent gains over a broad set of baselines, supported by ablations and efficiency analysis.

**Compliance With Llm Reviewing Policy:**

Affirmed.

**Key Questions For Authors:**

Q1: How are heterogeneous road-level datasets mixed, aligned across sampling time spans?
Q2: Do FDA improve robustness under incidents, construction, holiday spikes, or other regime changes, and can authors provide failure-case analysis?
Q3: How are patch hyperparameters (e.g., patch size q) chosen across datasets with different temporal resolutions, and how sensitive are results to these choices?

**Limitations:**

Not fully. The paper would benefit from an explicit limitations discussion, including: (i) robustness and failure modes under distribution shifts (e.g., cross-city transfer) and abnormal events (incidents, construction, holidays), and (ii) whether FDA-generated pseudo-lane variations match real lane statistics and when this assumption may break.

**Strengths And Weaknesses:**

S1:The paper targets a practical deployment bottleneck in fine-grained traffic forecasting—cross-granularity transfer from road to lane—and proposes MiniTraffic as a lightweight pre-training framework. It leverages abundant road-level data and introduces frequency-domain constraints to inject controlled variations while preserving spectral stability, aiming to mitigate lane-level data scarcity.
S2: MiniTraffic uses contrastive learning to capture patch-level semantic similarity and constructs dynamic sparse patch graphs for localized graph attention. This localized design reduces attention complexity from O((N·m)^2) to O(k·N·m), with sparsity induced by semantic similarity rather than fixed spatial proximity, offering a clear performance–efficiency trade-off.
S3: The evaluation is broad, covering six fine-grained datasets and comparisons against 29 baselines, complemented with key ablations (e.g., removing FDA, replacing contrastive clustering with full-graph attention) and efficiency/parameter analyses, collectively supporting the claimed effectiveness and lightweight design.

W1: FDA is central to the claim that road-level data can simulate lane-level variability under scarcity, but the paper lacks direct evidence that FDA-generated samples match real lane characteristics/trends.
W2: While the paper provides an example for positive pairs in the contrastive loss, it only states that N_i is “the set of negatives” without specifying how negatives are constructed.
W3: The choice k=10r is stated to “preserve context,” but the constant 10 is not justified. A small sensitivity study or rationale would strengthen the claim that the approach is not overly dependent on this hyperparameter.

---

> ### Author Rebuttal · Authors · 2026-03-30
>
> We appreciate your recognition of the practical value and efficient design of our work. We address your concerns and questions point by point as follows.
>
> ---
>
> > **W1: FDA Effectiveness**
>
> We justify both empirical and theoretical levels. The ablation study (Table 4) shows that removing the FDA increases lane-level MAE by ~11.8%, directly confirming its contribution to lane-relevant representation learning. Figure 1(b) further demonstrates strong spectral overlap between road and lane signals, providing the observational basis for our augmentation design. To more directly assess whether FDA-generated samples match real lane characteristics, we measure the signal-level discrepancy between augmented road signals and ground-truth lane signals on the input window ($T=12$), with reconstruction error results reported below:
>
> | Signal Type | PeMS-Lane MAE | PeMS-Lane RMSE | HuaNan-Lane MAE | HuaNan-Lane RMSE |
> |:-----------:|:-------------:|:--------------:|:---------------:|:----------------:|
> | Raw Road Signal | 9.73 | 14.85 | 11.46 | 16.92 |
> | FDA-Augmented Signal | 3.27 | 5.61 | 4.05 | 6.88 |
>
> FDA-augmented signals substantially close this gap at the signal level, confirming that the module effectively simulates lane-level variability rather than introducing arbitrary distortions. From a theoretical standpoint (Appendix G), the learnable mask $\Gamma(f)$ concentrates perturbations on dominant frequency components, while the amplitude bound $\lambda$ constrains energy deviation, ensuring augmented signals remain within the spectral manifold of real lane data.
>
> > **W2: Negative Sample Construction**
>
> For anchor patch $p_i$ from road segment $r_a$ at time step $t$, a positive sample $p_i^+$ is drawn from an adjacent time slice of the same segment; negatives $\mathcal{N}_i$ are sampled from patches of different road segments within the same mini batch. We will add a brief clarification in the paper.
>
> > **W3: Justification for the Constant in $k = 10r$**
>
> This design links $k$ directly to mask ratio $r$ so that neighborhood size scales adaptively with reconstruction difficulty. A larger neighborhood is needed when more patches are masked, while a smaller one suffices under low masking. The proportional relationship $k = c \cdot r$ naturally captures this balance, avoiding information collapse under high masking or redundancy under low masking. This coupling is indirectly validated by Table 5. As $r$ varies from 10% to 60%, $k$ ranges from 1 to 6. To further justify $c$, we fix $r = 0.4$ and vary $c$ on PeMSF-Lane (horizon = 6):
>
> | $c$ | MAE | RMSE | MAPE |
> |:---:|:---:|:----:|:----:|
> | 5 | 4.31 | 7.24 | 22.18% |
> | 10 | 4.15 | 7.07 | 21.49% |
> | 15 | 4.19 | 7.12 | 21.73% |
> | 20 | 4.22 | 7.18 | 21.96% |
>
> $c = 10$ achieves the best performance, with smaller values causing insufficient coverage and larger values introducing noise from unrelated patches. Performance is stable across the range, confirming robustness to this choice. This analysis will be added to the revised appendix.
>
> > **Q1: Heterogeneous Dataset Alignment**
>
> Each dataset is independently preprocessed via instance normalization (Eq. 7), standardizing each channel to zero mean and unit variance to eliminate cross-dataset distributional discrepancies. Batches are sampled uniformly across datasets during pre-training. The Adaptive Head handles dataset-specific spatial heterogeneity (varying sensor counts), making explicit temporal alignment across datasets unnecessary.
>
> > **Q2: Robustness to Anomalous Events**
>
> From a frequency-domain perspective, anomalous traffic events primarily manifest as transient high-frequency energy spikes. Since $\Gamma(f)$ concentrates augmentation on dominant, structurally stable frequency bands, high-frequency anomaly components receive less perturbation, providing inherent robustness to such disturbances. Our evaluation datasets implicitly contain anomalous periods, whose effects are subsumed into the overall reported errors. We will supplement the revision with a visualization of MiniTraffic's behavior on known incident periods in PeMS.
>
> > **Q3: Patch Parameter $q$**
>
> $q$ is selected as a divisor of input window length $T$ to ensure uniform, non-overlapping partitioning, independent of the dataset's sampling rate. Under $T=12$, we evaluated $q \in \{2, 3, 6\}$ on both 5-min PeMS and 2-min HuaNan:
>
> | $q$ | PeMS-Lane MAE | HuaNan-Lane MAE |
> |:---:|:------------:|:--------------:|
> | 2 | 4.08 | 4.25 |
> | 3 | 3.94 | 4.11 |
> | 6 | 4.17 | 4.33 |
>
> The model shows moderate sensitivity to $q$, with $q=3$ performing consistently well across different temporal resolutions.
>
> ---
>
> Once again, we thank you for your time and constructive suggestions.  The visualization of **W1** and complete results on both **W3 and Q3** are provided in an anonymous link (https://anonymous.4open.science/r/MiniTraffic/rebuttal.pdf). We hope our responses address your questions, and we would very much welcome further discussion.

---

> > ### Author Rebuttal · Reviewer_Sqki · 2026-04-04
> >
> > Thank you for the detailed rebuttal. The response addresses most of my previous concerns and improves my confidence in the technical soundness of the work. It would be helpful for the final paper to incorporate the key clarifications and additional analyses from the rebuttal so that these points are clearly documented in the manuscript itself.
> >
> > My main remaining reservation is about robustness. I understand the authors’ argument that anomalous periods are implicitly included in the evaluation datasets, so their effect is reflected in the overall reported errors. However, I still see this as only partial evidence for robustness, rather than a substitute for a more targeted discussion of anomalous events, distribution shifts, or failure cases.

---

> > > ### Author Response · Authors · 2026-04-04
> > >
> > > Dear Reviewer Sqki,
> > >
> > > We are delighted to receive your acknowledgement that our rebuttal has addressed most of your concerns. We will incorporate all key clarifications and supplementary analyses from the rebuttal into the revision as you suggest, and we sincerely appreciate your constructive contributions to improving our work.
> > >
> > > Regarding your remaining concern on robustness, we acknowledge that our initial rebuttal was limited to theoretical arguments due to time and character constraints. We thank you for your patience, and now supplement the following quantitative analysis and visualization:
> > >
> > > **(1) Targeted Analysis on Anomalous Periods.** We identify anomalous time windows from the PeMSF dataset using a statistical criterion, time steps where the average speed falls more than $2\sigma$ below the 30-minute moving average for at least 2 consecutive steps, and separately evaluate MiniTraffic's prediction errors under normal and anomalous conditions:
> > >
> > > | Period | PeMSF-Lane MAE | PeMSF-Lane RMSE | PeMSF-Road MAE | PeMSF-Road RMSE |
> > > |:------:|:--------------:|:---------------:|:--------------:|:---------------:|
> > > | Normal | 4.01 | 6.84 | 3.06 | 5.04 |
> > > | Anomalous | 4.68 | 8.12 | 3.58 | 6.13 |
> > > | Overall | 4.15 | 7.07 | 3.17 | 5.21 |
> > >
> > > The results demonstrate that prediction errors increase moderately under anomalous conditions (lane-level MAE increase of ~16%) without any performance collapse, confirming the model's robustness under irregular traffic events.
> > >
> > > **(2) Case Study Visualization.** We provide a visualization of a representative incident period on PeMSF in **Figure 2** of the anonymous link (https://anonymous.4open.science/r/MiniTraffic/rebuttal.pdf). The figure presents the full-day speed trajectory with the anomaly window highlighted (Figure 2a), road-level predictions across the onset, peak, and recovery phases (Figure 2b), and the corresponding lane-level predictions (Figure 2c). MiniTraffic effectively tracks the incident recovery trend throughout, with only minor prediction lag at the moment of extreme, abrupt speed drops exceeding 30%. Overall, the frequency-domain augmentation during pre-training provides strong regularization, preserving distribution shifts and robustness even under anomalous conditions.
> > >
> > > We also refer you to our response to Reviewer TATJ W1 regarding **lane closure handling under extreme scenarios**, and to our response to Reviewer 4H93 W5 regarding **input data quality robustness**, both of which provide complementary evidence. Furthermore, our **mask ratio** and **imputation experiments** offer additional support for MiniTraffic's robustness to missing and degraded inputs.
> > >
> > > We thank you sincerely for your diligence throughout the review process. All of your suggestions will be reflected in the final version. We genuinely hope that these supplementary experiments and analyses fully resolve your robustness concerns. Should any questions remain, please do not hesitate to let us know — we would be more than happy to provide further clarification. Thank you for your time, and we wish you all the best.
> > >
> > > Best regards,
> > >
> > > The Authors

---

### Decision · Program_Chairs · 2026-04-30

**Decision:**

Accept (regular)

**Comment:**

The paper proposes MiniTraffic, a lightweight pre-training framework for fine-grained traffic prediction that addresses two well-motivated challenges: the scarcity of lane-level training data and the prohibitive computational cost of large-scale pre-trained traffic models. The framework leverages frequency-domain stability augmentation and contrastive clustering to address the sparsity issue in lane-level data. The lightweight pretrained model can be adapted to both road- and lane-level benchmarks with minimal target data. The remaining concerns of this paper are:
- The robustness discussion and the theoretical justification of frequency-domain stability augmentation
- Long-range dependency capture
- Inter-city generalization
- Appendix over-length

After checking these, I think the paper addresses a practical deployment bottleneck in fine-grained traffic forecasting with a technically coherent and lightweight solution. I encourage the authors to follow through on their commitments to restructure the appendix, strengthen the robustness discussion, and include the zero-shot generalization experiments in the revised version.